# PROXY DENOISING FOR SOURCE-FREE DOMAIN ADAPTATION

**Song Tang[1,2,3], Wenxin Su[1], Yan Gan[4], Mao Ye[5,*], Jianwei Zhang[2] & Xiatian Zhu[6,*]**
[1]University of Shanghai for Science and Technology, [2]Universität Hamburg, [3]ComOriginMat Inc.
[4]Chongqing University, [5]University of Electronic Science and Technology of China, [6]University of Surrey
tangs@usst.edu.cn, {suwenxin43,cvlab.uestc}@gmail.com, xiatian.zhu@surrey.ac.uk

## ABSTRACT

Source-Free Domain Adaptation (SFDA) aims to adapt a pre-trained source model to an unlabeled target domain with no access to the source data. Inspired by the success of large Vision-Language (ViL) models in many applications, the latest research has validated ViL's benefit for SFDA by using their predictions as pseudo supervision. However, we observe that ViL's supervision could be noisy and inaccurate at an unknown rate, introducing additional negative effects during adaption. To address this thus-far ignored challenge, we introduce a novel *Proxy Denoising* (**ProDe**) approach. The key idea is to leverage the ViL model as a proxy to facilitate the adaptation process towards the latent domain-invariant space. We design a proxy denoising mechanism to correct ViL's predictions, grounded on a proxy confidence theory that models the dynamic effect of proxy's divergence against the domain-invariant space during adaptation. To capitalize on the corrected proxy, we derive a mutual knowledge distilling regularization. Extensive experiments show that ProDe significantly outperforms current state-of-the-art alternatives under the conventional closed set setting and more challenging open set, partial set, generalized SFDA, multi-target, multi-source, and test-time settings. Our code and data are available at https://github.com/tntek/source-free-domain-adaptation.

## 1 INTRODUCTION

Unsupervised Domain Adaptation (UDA) uses well-annotated source data and unannotated target data concurrently to achieve cross-domain transfer. However, this data access requirement raises increasing concerns about safety and privacy. Thus, there is a call for restricted access to source domain training data, leading to a more practical but challenging transfer learning setting, Source-Free Domain Adaptation (SFDA) (Li et al., 2020a; Xia et al., 2021; Roy et al., 2022).

In the absence from the source domain, cross-domain distribution matching approaches are no longer applicable (Ganin & Lempitsky, 2015; Kang et al., 2019). Self-supervised learning then comes into play by generating and mining auxiliary information to enable unsupervised adaptation in two main routes. *The first* makes SFDA as a special case of UDA by explicitly creating a pseudo source domain, enabling UDA methods such as adversarial learning (Xia et al., 2021; Kurmi et al., 2021) or minimizing domain shift (Tian et al., 2022; Kundu et al., 2022). *The second* further refines generated supervision from the source model (Lao et al., 2021; Wang et al., 2022a; Huang et al., 2021) or target data (Yang et al., 2022; Tang et al., 2022), as the constructed pseudo source domain may be noisy. These methods all perform alignment without any guidance from the target feature space to the unknown domain-invariant feature space.

There has been growing interest in leveraging pre-trained Vision-Language (ViL) models (e.g., CLIP (Radford et al., 2021)) for transfer learning challenges. This is because ViL models were trained with a massive amount of diverse vision-language data, encompassing rich knowledge potentially useful for many downstream tasks. For instance, Ge et al. (2022); Lai et al. (2023); Singha et al. (2023) disentangle domain and category information in the visual features of the ViL model

---
*Corresponding author

by learning domain-specific textual or visual prompts. ViL models have also been used to address the SFDA problem (Tang et al., 2024c; Xiao et al., 2024). They treat the ViL model's predictions as ground truth which would be noisy in many unknown cases, finally harming their performance.

To address the limitation mentioned above, in this paper, we propose a new **Pro**xy **De**noising (**ProDe**) approach for SFDA. In contrast to (Tang et al., 2024c; Xiao et al., 2024), we consider the ViL model/space as a *noisy* proxy of the latent domain-invariant space[1], with a need to be denoised. At the absence of any good reference models for measuring the noisy degree with the already strong ViL model's predictions, we exploit *the dynamics of domain adaptation process*, starting at the source model space and terminating presumably in the latent domain-invariant space. In particular, this takes into account the proxy's divergence against the domain-invariant space (Fig. 1). Specifically, we model approximately the effect of ViL model's prediction error on domain adaption by formulating a proxy confidence theory, in relation to the discrepancy between the source domain and the current under-adaptation model. This leads to a novel proxy denoising mechanism for ViL prediction correction. To capitalize on the corrected ViL predictions more effectively, a mutual knowledge distilling regularization is further designed.

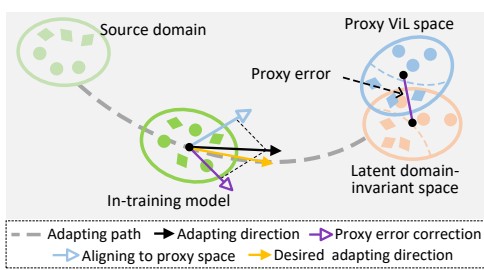

Figure 1: Conceptual illustration of ProDe. We align the adapting direction with the desired trajectory by leveraging a proxy space that approximates the latent domain-invariant space. This process incorporates direction adjustments based on proxy error correction, implementing proxy denoising, and finally achieves enhanced model adaptation.

Our **contributions** are summarized as follows: **(1)** We for the first time investigate the inaccurate predictions of ViL models in the context of SFDA. **(2)** We formulate a novel ProDe method that reliably corrects the ViL model's predictions under the guidance of a proxy confidence theory. A mutual knowledge distilling regularization is introduced for better capitalizing on refined proxy predictions. **(3)** Extensive experiments on open benchmarks show that our ProDe significantly outperforms previous alternatives in closed-set settings, as well as the more challenging partial-set, open-set, and generalized SFDA, multi-target, multi-source and test-time settings.

## 2 RELATED WORK

**Source-Free Domain Adaptation** One main challenge with SFDA is lack of supervision during model adaptation. To overcome this, current methods are broadly divided into three categories. The *first category* involves converting SFDA to conventional UDA by introducing a pseudo-source domain. This can be achieved by building the pseudo-source domain through generative models (Tian et al., 2022; Li et al., 2020b) or by extracting a subset similar to the distribution of sources from the target domain (Du et al., 2023). The *second category* involves mining auxiliary information from the pre-trained source model to assist in aligning the feature distribution from the target domain to the source domain. Commonly used auxiliary factors include multi-hypothesis (Lao et al., 2021), prototypes (Zhou et al., 2024), source distribution estimation (Ding et al., 2022), or hard samples (Li et al., 2021). The *last category* focuses on the target domain and creates additional constraints to correct the semantic noise in model transferring. In practice, domain-aware gradient control (Yang et al., 2021b), data geometry such as the intrinsic neighborhood structure (Tang et al., 2021) and target data manifold (Tang et al., 2022; Tang et al., 2024a), have been exploited to generate high-quality pseudo-labels (Liang et al., 2020; Chen et al., 2022b) or inject assistance in an unsupervised fashion (Yang et al., 2021a). These methods refine auxiliary information from domain-specific knowledge, such as the source model and unlabeled target data, without resorting to external knowledge sources, such as pre-trained multimodal foundation models.

---

[1]The issue of noisy predictions is evidenced by the inferior zero-shot performance of the ViL model, e.g., CLIP, on the target domains (see Tab. 4). Here, "domain invariant space" refers to an ideal latent embedding space where the mapped features from different domains align with the same probability distribution.

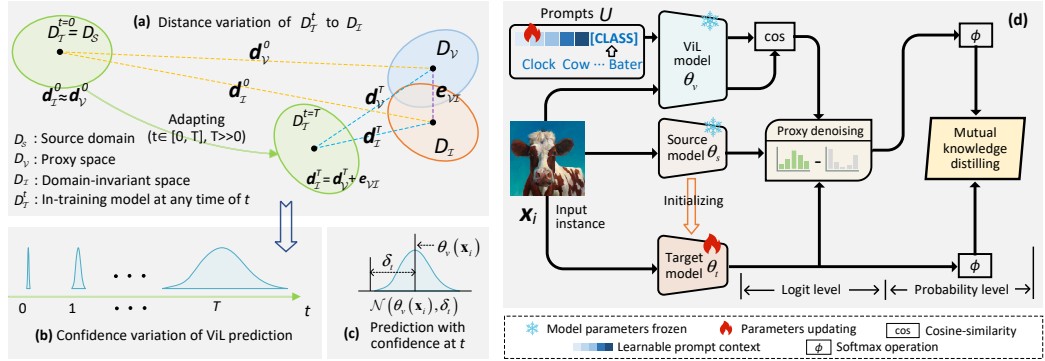

Figure 2: **Left:** Dynamics of effect of ViL model's prediction error (or proxy error) during alignment. (a) In the initial adaptation phase, it is acceptable to overlook the proxy errors. However, as the in-training model approaches the proxy space, these errors grow to be more noticeable, leading to continuous decline in the reliability of ViL predictions as shown in (b) and (c). **Right:** Our ProDe capitalizes on the corrected proxy, involving a mutual knowledge distilling regularization and a proxy denoising mechanism imposing refinement on the ViL logits.

**Vision-Language Models**  ViL models, such as CLIP (Radford et al., 2021) and GLIP (Li et al., 2022), have shown promise in various tasks (Liang et al., 2023; Wang et al., 2022c) due to their ability to capture modality invariant features. There are two main lines of research. The *first line* aims to improve their performance. For instance, text-prompt learning (Zhou et al., 2022; Ge et al., 2022) and visual-prompt learning (Wang et al., 2023; Jia et al., 2022) were adopted, using learnable prompts related to application scenarios. Data efficiency of these models can be improved by re-purposing (Andonian et al., 2022) or removing noisy data (Wang et al., 2021b). The *second line* focuses on using ViL models as external knowledge to boost downstream tasks. Three strategies are involved: Plain fusion (Liu et al., 2024), knowledge distillation (Pei et al., 2023) and information entropy regulating (Cha et al., 2022). Beyond latest ViL based SFDA models (Tang et al., 2024c; Xiao et al., 2024), we uniquely tackle the challenge of mitigating the noise of ViL's supervision.

## 3 METHODOLOGY

### 3.1 PROBLEM FORMULATION

We start with a labeled source domain and an unlabeled target domain, handling the same $C$ categories. Let $\mathcal{X}_\mathcal{S}$ and $\mathcal{Y}_\mathcal{S}$ be the source samples and labels. The target samples and truth target labels are denoted as $\mathcal{X}_\mathcal{T} = \{\boldsymbol{x}_i\}_{i=1}^n$ and $\mathcal{Y}_\mathcal{T} = \{y_i\}_{i=1}^n$, respectively, where $n$ is the sample number. SFDA aims to learn a target model $\theta_t : \mathcal{X}_\mathcal{T} \to \mathcal{Y}_\mathcal{T}$ given (1) the pre-trained source model $\theta_s : \mathcal{X}_\mathcal{S} \to \mathcal{Y}_\mathcal{S}$, (2) the unlabeled target data $\mathcal{X}_\mathcal{T}$. In addition, we leverage a ViL model $\theta_v$ that produces noise supervision.

To address noisy ViL supervision, we exploit the dynamics of domain adaptation process. As shown in Fig. 2 (a), we deal with three spaces: source domain $D_\mathcal{S}$ (i.e., source image embedding space), domain-invariant space $D_\mathcal{I}$, and ViL space $D_\mathcal{V}$ (the best possible proxy w.r.t $D_\mathcal{I}$). In this context, $D_\mathcal{I}$ typically refers to an *ideal, unknown latent embedding space* that is domain generalized. We want to align the in-training model $D_\mathcal{T}^t$ from $D_\mathcal{S}$ to $D_\mathcal{I}$ as $t \in [0 \sim T] \gg 0$.

Without access to $D_\mathcal{I}$, we propose to perform ***proxy alignment*** by aligning $D_\mathcal{T}^t$ towards $D_\mathcal{V}$. We denote the discrepancy between $D_\mathcal{I}$ and $D_\mathcal{V}$ as ***proxy error*** $\boldsymbol{e}_{\mathcal{V}\mathcal{I}}$, reflecting ViL's prediction errors. We then transform the task of minimizing the errors of ViL predictions to control the proxy error by establishing a proxy confidence theory.

### 3.2 PROXY CONFIDENCE THEORY

Understanding the impact of proxy errors on domain adaptation is critical. To account for the dynamics of domain adaptation, as demonstrated in Fig. 2 (a), we consider two typical situations of the proxy alignment process. We denote the distance of $D_\mathcal{T}^t$ to $D_\mathcal{V}$ and $D_\mathcal{I}$ as $\boldsymbol{d}_\mathcal{V}^t$ and $\boldsymbol{d}_\mathcal{I}^t$,

respectively, and note that the distinction between $D_{\mathcal{V}}$ and $D_{\mathcal{I}}$, i.e., the proxy error $\boldsymbol{e}_{\mathcal{VI}}$, is a space-to-space distance in the vector form. To ease understanding, we note two cases:

- **Case1:** When $D_{\mathcal{T}}^t$ is way far from $D_{\mathcal{V}}$, e.g., at the beginning of adaptation ($t = 0$), it is held that $\boldsymbol{d}_{\mathcal{I}}^0 \approx \boldsymbol{d}_{\mathcal{V}}^0 \gg \boldsymbol{e}_{\mathcal{VI}}$. This implies that aligning to $D_{\mathcal{I}}$ or $D_{\mathcal{V}}$ is equivalent. Consequently, the proxy errors $\boldsymbol{e}_{\mathcal{VI}}$ can be ignored, that is, the ViL prediction can be deemed trustworthy.

- **Case2:** When $D_{\mathcal{T}}^t$ approaches $D_{\mathcal{V}}$, e.g., the later phase in the adaptation ($t = U \gg 0$), tackling the proxy errors becomes increasingly crucial; Also, the distance relationship evolves to the equation that $\boldsymbol{d}_{\mathcal{I}}^U = \boldsymbol{d}_{\mathcal{V}}^U + \boldsymbol{e}_{\mathcal{VI}}$ (according to the vector geometric property that $\boldsymbol{u}$, $\boldsymbol{v}$, and $\boldsymbol{u} + \boldsymbol{v}$ form a triangle, where $\boldsymbol{u}$ and $\boldsymbol{v}$ are two sides). At this moment, ViL predictions become less reliable.

The proxy errors dynamically affect the proxy alignment, as reflected in the relative relationship between $\boldsymbol{d}_{\mathcal{V}}^t$ and $\boldsymbol{d}_{\mathcal{I}}^t$ defined as:

$$\eta_t = \frac{|\boldsymbol{d}_{\mathcal{I}}^t|}{|\boldsymbol{d}_{\mathcal{V}}^t|} = \frac{|\boldsymbol{d}_{\mathcal{V}}^t + \boldsymbol{e}_{\mathcal{VI}}|}{|\boldsymbol{d}_{\mathcal{V}}^t|} \leq \frac{|\boldsymbol{d}_{\mathcal{V}}^t| + |\boldsymbol{e}_{\mathcal{VI}}|}{|\boldsymbol{d}_{\mathcal{V}}^t|} = 1 + \frac{|\boldsymbol{e}_{\mathcal{VI}}|}{|\boldsymbol{d}_{\mathcal{V}}^t|}, \tag{1}$$

where $\eta_t$ quantifies the *error impact degree*, $|\cdot|$ means the absolute value (length) of a distance vector. During proxy alignment, the quantity $|\boldsymbol{e}_{\mathcal{VI}}|/|\boldsymbol{d}_{\mathcal{V}}^t|$ in Eq. (1) gradually increases from a very small value (e.g., Case 1) to bigger ones (e.g., Case 2), leading to increase in impact degree $\eta_t$ from 1. With this dynamics, as shown in Fig. 2 (b), the variance of ViL prediction gradually increases, implying a progressive decrease in the reliability of ViL prediction.

At any time $t$, we treat the ViL predictions that approximate a Gaussian distribution $\mathcal{N}(\theta_v(x_i), \delta_t)$ with the mean $\theta_v(x_i)$ and the prediction variance $\delta_t \propto \eta_t$ (Fig. 2 (c)). This is because, we consider the ViL predictions to be influenced by various sources of noise and uncertainty, which justifies the Gaussian approximation according to the *Central Limit Theorem* (Chow & Teicher, 1988).

Given that $\boldsymbol{e}_{\mathcal{VI}}$ is unknown, we cannot formulate these dynamics explicitly. We thus approximate this problem by quantifying the prediction variance with the varying confidence of ViL predictions. This conversion can be expressed in the form of a probability distribution with proxy confidence as:

$$\mathcal{N}(\theta_v(x_i), \delta_t) \Longrightarrow P(G_{P(\mathcal{V})} = True, t) P(\mathcal{V}), \tag{2}$$

where $P(\mathcal{V})$ is the probability distribution of the proxy space $D_{\mathcal{V}}$; $G_{P(\mathcal{V})}$ stands for a random event that the sampling result (i.e., a ViL prediction) from $P(\mathcal{V})$ is confident; $P(G_{P(\mathcal{V})} = True, t)$ is denoted as the *proxy confidence*, indicating the probability of the event $G_{P(\mathcal{V})}$ being true at a time $t$. This confidence will decreases progressively, as the ViL prediction reliability reduces relatively against the ability of the in-training model.

By framing the ViL prediction as a probabilistic event, we can leverage the concept of proxy confidence, $P(G_{P(V)} = True, t)$, to quantify the reliability of ViL predictions at any point during adaptation. This facilitates the measurement about the impact of proxy errors. Specifically, we formulate the *proxy confidence theory* as in **Theorem 1** (see proof in `Appendix A`).

**Theorem 1** *We note that the source domain ($D_{\mathcal{S}}$), the domain-invariant space ($D_{\mathcal{I}}$), the proxy space ($D_{\mathcal{V}}$) and the in-training model ($D_{\mathcal{T}}^t$) follow the probability distributions $P(\mathcal{S})$, $P(\mathcal{I})$, $P(\mathcal{V})$ and $P(\mathcal{T}^t)$, respectively, where $\mathcal{S}$, $\mathcal{I}$, $\mathcal{V}$ and $\mathcal{T}^t$ are corresponding random variables. With our proxy alignment idea (see `Sec. 3.1`), the proxy confidence can be expressed as:*

$$P(G_{P(\mathcal{V})} = True, t) \propto \frac{P(\mathcal{T}^t)}{P(\mathcal{S})}. \tag{3}$$

This theorem tells that *the effect of ViL prediction errors on domain adaption can be approximately estimated by contrasting the distributions of the source model and the current in-training model.*

### 3.3 CAPITALIZING ON THE CORRECTED PROXY

**Overview** To better leverage the corrected proxy, we propose a novel ProDe method featured with two designs: (1) A proxy denoising mechanism, refining the original ViL predictions at the logit level, and (2) a mutual knowledge distilling regularization, encouraging extraction of useful knowledge from the ViL model $\theta_v$ to the in-training target model $\theta_t$, as shown in Fig. 2 (d).

**Proxy denoising** This module aims to denoise the ViL predictions. By **Theorem** 1 (Eq. (3)), we further convert the ViL space's probability distribution with proxy confidence (i.e., Eq. (2)) into

$$\log \left( \frac{P(\mathcal{T}^t)}{P(\mathcal{S})} P(\mathcal{V}) \right) = \log P(\mathcal{V}) - \left[ \log P(\mathcal{S}) - \log P(\mathcal{T}^t) \right], \tag{4}$$

where the latter two items form an adjustment used to correct for the first item (i.e., ViL prediction). Under this formula, we realize our denoising mechanism as:

$$\boldsymbol{p}'_i = \mathrm{softmax} \left( \theta_v \left( \boldsymbol{x}_i, \boldsymbol{v} \right) - \omega[\theta_s \left( \boldsymbol{x}_i \right) - \theta_t \left( \boldsymbol{x}_i \right)] \right), \tag{5}$$

where $\theta_v/\theta_s/\theta_t()$ apply the ViL/source/target model to get the corresponding logits, and the hyperparameter $\omega$ specifies the correction strength. The output $\boldsymbol{p}'_i$ is a denoised ViL prediction.

**Mutual knowledge distilling** This component aims to distill useful knowledge of the ViL model to our target model. This is achieved by designing two loss terms:

$$L_{\mathrm{ProDe}} = \min_{\theta_t, \boldsymbol{v}} \alpha \overbrace{\left( -\mathbb{E}_{\boldsymbol{x}_i \in \mathcal{X}_t} \mathbf{MI} \left( \boldsymbol{p}'_i, \boldsymbol{p}_i \right) + \gamma \sum_{c=1}^{C} \bar{q}_c \log \bar{q}_c \right)}^{L_{\mathrm{Apt}}} - \beta \overbrace{\mathbb{E}_{\boldsymbol{x}_i \in \mathcal{X}_t} \sum_{c=1}^{C} \mathbb{1} \left[ c = y'_i \right] \log p_{i,c}}^{L_{\mathrm{Ref}}}, \tag{6}$$

The first term $L_{\mathrm{Apt}}$ adapts both the target model and the learnable prompt of ViL model by maximizing the unbiased mutual information $\mathbf{MI}(\cdot, \cdot)$ (Ji et al., 2019) between the denoised ViL prediction $\boldsymbol{p}'_i$ and the target prediction $\boldsymbol{p}_i = \mathrm{softmax}(\theta_t(\boldsymbol{x}_i))$. This design is motivated by that despite massive (often noisy) training data used, the ViL model (e.g., CLIP) don't always outperform a speical expert model such as the supervised source model. There are three reasons: (1) ViL models are generalists, while source domain models are specialized. (2) ViL models may include irrelevant data, whereas source domain models use curated, relevant data. (3) ViL models might overlook task-specific features that are captured by source domain models. To avoid solution collapse (Ghasedi Dizaji et al., 2017), we use a common category balance constraint (Yang et al., 2021a) where $\bar{q}_c = \frac{1}{n} \sum_{i=1}^{n} p_{i,c}$ is the average likelihood of class $c$ over $n$ training samples by the target model, across a total of $C$ categories.

The second term $L_{\mathrm{Ref}}$ refers to a typical pseudo labeling strategy where a classification objective is applied, with the pseudo label $y'_i$ obtained by the denoised ViL predictions and $\mathbb{1}[c = y'_i]$ denotes an indicator function. Note that as the training proceeds, the ViL predictions become less reliable and useful whilst the negative effect of $\boldsymbol{e}_{\mathcal{V}\mathcal{I}}$ would grow in a relative sense. That means our proposed denoising could get more important across adaptation.

We provide the model training procedure in `Appendix B`.

## 4 EXPERIMENTS

**Datasets** We evaluate four widely used domain adaptation benchmarks. Among them, **Office-31** (Saenko et al., 2010) and **Office-Home** (Venkateswara et al., 2017) are small-scaled and medium-scale datasets, respectively, whilst **VisDA** (Peng et al., 2017) and **DomainNet-126** (Saito et al., 2019) are both challenging large-scale datasets. Their details are provided in `Appendix C`.

**Settings** We consider a variety of SFDA settings: (1) closed-set, (2) partial-set (initialized in SHOT (Liang et al., 2020)), (3) open-set (initialized in SHOT (Liang et al., 2020)), (4) generalized SFDA (Yang et al., 2021b), (5) multi-target (SF-MTDA, detailed in (Kumar et al., 2023)), (6) multi-source (SF-MSDA, detailed in (Ahmed et al., 2021)), and (7) test-time adaptation (TTA) (Wang et al., 2021a). More details are given in `Appendix D`.

### 4.1 COMPETITORS

To evaluate ProDe, we select 30 related comparisons divided into four groups. *(1) The first* includes 2 base models involved in the SFDA problem: The source model (termed Source) and CLIP zero-shot (termed CLIP) (Radford et al., 2021). *(2) The second* includes 7 current state-of-the-art domain adaptation methods with ViL model (adopting CLIP in practice), covering UDA and SFDA settings: DAPL-R (Ge et al., 2022), PADCLIP-R (Lai et al., 2023), ADCLIP-R (Singha et al., 2023), PDA-R (Bai et al., 2024), DAMP-R (Du et al., 2024), DIFO-R (Tang et al., 2024c) and DIFO-V (Tang

et al., 2024c). Among them, DIFO-R and DIFO-V are the SFDA methods, while others are UDA methods. The suffix of "-R" and "-V" means that the image-encoder in CLIP uses the backbone of ResNet and ViT, respectively. Specifically, DIFO-V employs the backbone of ViT-B/32 across all datasets, whilst the rest methods with "-R" use ResNet101 on VisDA and ResNet50 on the other three datasets. *(3) The third* comprises 16 state-of-the-art SFDA models without using ViL model: SHOT (Liang et al., 2020), NRC (Yang et al., 2021a), GKD (Tang et al., 2021), HCL (Huang et al., 2021), AaD (Yang et al., 2022), AdaCon (Chen et al., 2022a), CoWA (Lee et al., 2022), ELR (Yi et al., 2023), PLUE (Litrico et al., 2023), CRS (Zhang et al., 2023), CPD (Zhou et al., 2024), TPDS (Tang et al., 2024a), GDA (Yang et al., 2021b), PSAT-ViT (Tang et al., 2024b) CoNMix (Kumar et al., 2023) and DECISION (Ahmed et al., 2021). Among them, GDA and PSAT-ViT are specific for the generalized SFDA setting, while CoNMix and DECISION are SF-MTDA and SF-MSDA methods, respectively. *(4) The fourth* comprises 5 state-of-the-art TTA models: Tent (Wang et al., 2021a), T3A (Iwasawa & Matsuo, 2021), CoTTA (Wang et al., 2022b), EATA (Niu et al., 2022) and SAR (Niu et al., 2023). Additionally, for a fair comparison with DIFO, the previous best SFDA method with ViL model, we have initiated ProDe into the same versions mentioned above: A strong version ProDe-V and a weak version ProDe-R.

## 4.2 COMPARATIVE EVALUATIONS

**Closed-set SFDA.** Tab. 1∼3 lists the quantitative comparisons on the four evaluation datasets. Both ProDe-R and ProDe-V beat all non-multimodal SFDA methods by a large margin. Compared with the second-best method CPD (Office-31), TPDS (Office-Home), PLUE (VisDA) and GKD (DomainNet-126), ProDe-V improves by **1.8%**, **11.0% 2.7%** and **16.3%** in average accuracy, respectively. As for those methods with CLIP, ProDe also beat them in the same backbone setting. In particular, compared with the multimodal SFDA method DIFO, ProDe improves by **4.8%** and **5.0%** (DomainNet-126) at most using ResNet and ViT-B/32, respectively. Actually, the weak version of our method, ProDe-R, is competitive with the strong version of DIFO, DIFO-V. All of these results indicate that ProDe can significantly boost the cross-domain adaptation under the SFDA setting.

Table 1: Closed-set SFDA results (%) on **Office-31**. **SF** means source-free.

| Method | Venue | SF | A→D | A→W | D→A | D→W | W→A | W→D | Avg. |
|---|---|---|---|---|---|---|---|---|---|
| Source | – | – | 79.1 | 76.6 | 59.9 | 95.5 | 61.4 | 98.8 | 78.6 |
| SHOT | ICML20 | ✓ | 93.7 | 91.1 | 74.2 | 98.2 | 74.6 | 100. | 88.6 |
| NRC | NIPS21 | ✓ | 96.0 | 90.8 | 75.3 | 99.0 | 75.0 | 100. | 89.4 |
| GKD | IROS21 | ✓ | 94.6 | 91.6 | 75.1 | 98.7 | 75.1 | 100. | 89.2 |
| HCL | NIPS21 | ✓ | 94.7 | 92.5 | 75.9 | 98.2 | 77.7 | 100. | 89.8 |
| AaD | NIPS22 | ✓ | 96.4 | 92.1 | 75.0 | 99.1 | 76.5 | 100. | 89.9 |
| AdaCon | CVPR22 | ✓ | 87.7 | 83.1 | 73.7 | 91.3 | 77.6 | 72.8 | 81.0 |
| CoWA | ICML22 | ✓ | 94.4 | 95.2 | 76.2 | 98.5 | 77.6 | 99.8 | 90.3 |
| ELR | ICLR23 | ✓ | 93.8 | 93.3 | 76.2 | 98.0 | 76.9 | 100. | 89.6 |
| PLUE | CVPR23 | ✓ | 89.2 | 88.4 | 72.8 | 97.1 | 69.6 | 97.9 | 85.8 |
| CPD | PR24 | ✓ | 96.6 | 94.2 | 77.3 | 98.2 | 78.3 | 100. | 90.8 |
| TPDS | IJCV24 | ✓ | 97.1 | 94.5 | 75.7 | 98.7 | 75.5 | 99.8 | 90.2 |
| DIFO-R | CVPR24 | ✓ | 93.6 | 92.1 | 78.5 | 95.7 | 78.8 | 97.0 | 89.3 |
| DIFO-V | CVPR24 | ✓ | 97.2 | 95.5 | 83.0 | 97.2 | 83.2 | 98.8 | 92.5 |
| **ProDe-R** | – | ✓ | 94.4 | 92.1 | 79.8 | 95.6 | 79.0 | 98.6 | 89.9 |
| **ProDe-V** | – | ✓ | 96.8 | 96.4 | 83.1 | 97.0 | 82.5 | 99.8 | 92.6 |

*Comparison to the ViL model.* We conducted a quantitative comparison between our model and CLIP's zero-shot performance. The results of our model are reported with average accuracy. As reported in Tab. 4, ProDe-R and ProDe-V improve at least by **5.0** % (on VisDA) and **8.1**% (on VisDA), respectively, compared with CLIP's results on the four datasets. This result shows that *the multimodal CLIP space only approximates the domain-invariant space, suggesting the need for denoising that this paper focuses on*.

**Partial-set and open-set SFDA.** For a complete evaluation, we also evaluate ProDe on two variation scenarios: Partial-set and open-set settings. As reported in Tab. 5, ProDe-V achieves a gain of **0.1**% (partial-set) and **6.7**% (open-set) compared with the best competitor DIFO-V.

**Generalized SFDA.** The generalized SFDA is an extended problem of closed-set SFDA, highlighting the anti-forgetting ability on the seen source domain. The same as (Yang et al., 2021b), we adopt the harmonic mean accuracy as evaluation protocol, which is computed by $H = (2 * Acc_s * Acc_t)/(Acc_s + Acc_t)$ where $Acc_s$ and $Acc_t$ are the accuracies of the adapted target model on the source domain and the target domain, respectively. Note that the $Acc_s$ is computed based on the source-testing set. The same to (Yang et al., 2021b; Tang et al., 2024b), on the source domain, the ratio of training and testing sets is 9:1. To evaluate effectiveness, two generalized SFDA methods, GDA and PSAT-ViT, are chosen as additional comparisons. Based on Tab. 6, it is seen that ProDe-V outperforms all comparisons in terms of H-accuracy besides PSAT-ViT with anti-forgetting

Table 2: Closed-set SFDA results (%) on **Office-Home** and **VisDA**. **SF** means source-free. The full results on **VisDA** are provided in `Appendix E.1`.

| Method | Venue | SF | Ar→Cl | Ar→Pr | Ar→Rw | Cl→Ar | Cl→Pr | Cl→Rw | Pr→Ar | Pr→Cl | Pr→Rw | Rw→Ar | Rw→Cl | Rw→Pr | Avg. | Sy→Re |
|---|---|---|---|---|---|---|---|---|---|---|---|---|---|---|---|---|
| Source | – | – | 43.7 | 67.0 | 73.9 | 49.9 | 60.1 | 62.5 | 51.7 | 40.9 | 72.6 | 64.2 | 46.3 | 78.1 | 59.2 | 49.2 |
| SHOT | ICML20 | ✓ | 56.7 | 77.9 | 80.6 | 68.0 | 78.0 | 79.4 | 67.9 | 54.5 | 82.3 | 74.2 | 58.6 | 84.5 | 71.9 | 82.7 |
| NRC | NIPS21 | ✓ | 57.7 | 80.3 | 82.0 | 68.1 | 79.8 | 78.6 | 65.3 | 56.4 | 83.0 | 71.0 | 58.6 | 85.6 | 72.2 | 85.9 |
| GKD | IROS21 | ✓ | 56.5 | 78.2 | 81.8 | 68.7 | 78.9 | 79.1 | 67.6 | 54.8 | 82.6 | 74.4 | 58.5 | 84.8 | 72.2 | 83.0 |
| AaD | NIPS22 | ✓ | 59.3 | 79.3 | 82.1 | 68.9 | 79.8 | 79.5 | 67.2 | 57.4 | 83.1 | 72.1 | 58.5 | 85.4 | 72.7 | 88.0 |
| AdaCon | CVPR22 | ✓ | 47.2 | 75.1 | 75.5 | 60.7 | 73.3 | 73.2 | 60.2 | 45.2 | 76.6 | 65.6 | 48.3 | 79.1 | 65.0 | 86.8 |
| CoWA | ICML22 | ✓ | 56.9 | 78.4 | 81.0 | 69.1 | 80.0 | 79.9 | 67.7 | 57.2 | 82.4 | 72.8 | 60.5 | 84.5 | 72.5 | 86.9 |
| ELR | ICLR23 | ✓ | 58.4 | 78.7 | 81.5 | 69.2 | 79.5 | 79.3 | 66.3 | 58.0 | 82.6 | 73.4 | 59.8 | 85.1 | 72.6 | 85.8 |
| PLUE | CVPR23 | ✓ | 49.1 | 73.5 | 78.2 | 62.9 | 73.5 | 74.5 | 62.2 | 48.3 | 78.6 | 68.6 | 51.8 | 81.5 | 66.9 | 88.3 |
| CPD | PR24 | ✓ | 59.1 | 79.0 | 82.4 | 68.5 | 79.7 | 79.5 | 67.9 | 57.9 | 82.8 | 73.8 | 61.2 | 84.6 | 73.0 | 85.8 |
| TPDS | IJCV24 | ✓ | 59.3 | 80.3 | 82.1 | 70.6 | 79.4 | 80.9 | 69.8 | 56.8 | 82.1 | 74.5 | 61.2 | 85.3 | 73.5 | 87.6 |
| DAPL-R | TNNLS23 | ✗ | 54.1 | 84.3 | 84.8 | 74.4 | 83.7 | 85.0 | 74.5 | 54.6 | 84.8 | 75.2 | 54.7 | 83.8 | 74.5 | 86.9 |
| PADCLIP-R | ICCV23 | ✗ | 57.5 | 84.0 | 83.8 | 77.8 | 85.5 | 84.7 | 76.3 | 59.2 | 85.4 | 78.1 | 60.2 | 86.7 | 76.6 | 88.5 |
| ADCLIP-R | ICCVW23 | ✗ | 55.4 | 85.2 | 85.6 | 76.1 | 85.8 | 86.2 | 76.7 | 56.1 | 85.4 | 76.8 | 56.1 | 85.5 | 75.9 | 87.7 |
| PDA-R | AAAI24 | ✗ | 55.4 | 85.1 | 85.8 | 75.2 | 85.2 | 85.2 | 74.2 | 55.2 | 85.8 | 74.7 | 55.8 | 86.3 | 75.3 | 86.4 |
| DAMP-R | CVPR24 | ✗ | 59.7 | 88.5 | 86.8 | 76.6 | 88.9 | 87.0 | 76.3 | 59.6 | 87.1 | 77.0 | 61.0 | 89.9 | 78.2 | 88.4 |
| DIFO-R | CVPR24 | ✓ | 62.6 | 87.5 | 87.1 | 79.5 | 87.9 | 87.4 | 78.3 | 63.4 | 88.1 | 80.0 | 63.3 | 87.7 | 79.4 | 88.6 |
| DIFO-V | CVPR24 | ✓ | 70.6 | 90.6 | 88.8 | 82.5 | 90.6 | 88.8 | 80.9 | 70.1 | 88.9 | 83.4 | 70.5 | 91.2 | 83.1 | 90.3 |
| **ProDe-R** | – | ✓ | 64.0 | 90.0 | 88.3 | 81.1 | 90.1 | 88.6 | 79.8 | 65.4 | 89.0 | 80.9 | 65.5 | 90.2 | 81.1 | 88.7 |
| **ProDe-V** | – | ✓ | 72.7 | 92.3 | 90.5 | 82.5 | 91.5 | 90.7 | 82.5 | 72.5 | 90.8 | 83.0 | 72.6 | 92.2 | 84.5 | 91.0 |

Table 3: Closed-set SFDA results (%) on **DomainNet-126**. **SF** means source-free.

| Method | Venue | SF | C→P | C→R | C→S | P→C | P→R | P→S | R→C | R→P | R→S | S→C | S→P | S→R | Avg. |
|---|---|---|---|---|---|---|---|---|---|---|---|---|---|---|---|
| Source | – | – | 44.6 | 59.8 | 47.5 | 53.3 | 75.3 | 46.2 | 55.3 | 62.7 | 46.4 | 55.1 | 50.7 | 59.5 | 54.7 |
| SHOT | ICML20 | ✓ | 63.5 | 78.2 | 59.5 | 67.9 | 81.3 | 61.7 | 67.7 | 67.6 | 57.8 | 70.2 | 64.0 | 78.0 | 68.1 |
| GKD | IROS21 | ✓ | 61.4 | 77.4 | 60.3 | 69.6 | 81.4 | 63.2 | 68.3 | 68.4 | 59.5 | 71.5 | 65.2 | 77.6 | 68.7 |
| NRC | NIPS21 | ✓ | 62.6 | 77.1 | 58.3 | 62.9 | 81.3 | 60.7 | 64.7 | 69.4 | 58.7 | 69.4 | 65.8 | 78.7 | 67.5 |
| AdaCon | CVPR22 | ✓ | 60.8 | 74.8 | 55.9 | 62.2 | 78.3 | 58.2 | 63.1 | 68.1 | 55.6 | 67.1 | 66.0 | 75.4 | 65.4 |
| CoWA | ICML22 | ✓ | 64.6 | 80.6 | 60.6 | 66.2 | 79.8 | 60.8 | 69.0 | 67.2 | 60.0 | 69.0 | 65.8 | 79.9 | 68.6 |
| PLUE | CVPR23 | ✓ | 59.8 | 74.0 | 56.0 | 61.6 | 78.5 | 57.9 | 61.6 | 65.9 | 53.8 | 67.5 | 64.3 | 76.0 | 64.7 |
| TPDS | IJCV24 | ✓ | 62.9 | 77.1 | 59.8 | 65.6 | 79.0 | 61.5 | 66.4 | 67.0 | 58.2 | 68.6 | 64.3 | 75.3 | 67.1 |
| DAPL-R | TNNLS23 | ✗ | 72.4 | 87.6 | 65.9 | 72.7 | 87.6 | 65.6 | 73.2 | 72.4 | 66.2 | 73.8 | 72.9 | 87.8 | 74.8 |
| ADCLIP-R | ICCVW23 | ✗ | 71.7 | 88.1 | 66.0 | 73.2 | 86.9 | 65.2 | 73.6 | 73.0 | 68.4 | 72.3 | 74.2 | 89.3 | 75.2 |
| DAMP-R | CVPR24 | ✗ | 76.7 | 88.5 | 71.7 | 74.2 | 88.7 | 70.8 | 74.4 | 75.7 | 70.5 | 74.9 | 76.1 | 88.2 | 77.5 |
| DIFO-R | CVPR24 | ✓ | 73.8 | 89.0 | 69.4 | 74.0 | 88.7 | 70.1 | 74.8 | 74.6 | 69.6 | 74.7 | 74.3 | 88.0 | 76.7 |
| DIFO-V | CVPR24 | ✓ | 76.6 | 87.2 | 74.9 | 80.0 | 87.4 | 75.6 | 80.8 | 77.3 | 75.5 | 80.5 | 76.7 | 87.3 | 80.0 |
| **ProDe-R** | – | ✓ | 79.3 | 91.0 | 75.3 | 80.0 | 90.9 | 75.6 | 80.4 | 78.9 | 75.4 | 80.4 | 79.2 | 91.0 | 81.5 |
| **ProDe-V** | – | ✓ | 83.2 | 92.4 | 79.0 | 85.0 | 92.3 | 79.3 | 85.5 | 83.1 | 79.1 | 85.5 | 83.4 | 92.4 | 85.0 |

design (by a tiny gap of **0.2**%). Meanwhile, both ProDe-R and ProDe-V deliver balanced results across the source and target domains. This is due to the correction in the proxy denoising, which incorporates information from the source model, thereby mitigating forgetting of the source domain.

**SF-MTDA, SF-MSDA and TTA.** This part evaluates ProDe in broader SF-MTDA, SF-MSDA and TTA settings. For SF-MTDA, we treat multiple target domains as a single integrated domain and adapt the source model accordingly. For SF-MSDA, we follow the ensembling approach from (Ahmed et al., 2021), passing the target data through each adapted source model and averaging the soft predictions to derive the test labels. The results, as shown in the left side of Tab. 7, demonstrate that ProDe substantially outperforms state-of-the-art alternatives in both settings.

The right side of Tab. 7 reports the results on the online SFDA setting of TTA, where all comparison methods maintain a fixed batch size of 64, similar to ours. It is seen that ProDe demonstrates advantages over previous state-of-the-art methods.

### 4.3 MODEL ANALYSIS

**Feature distribution visualization.** Based on the task Cl→Ar in the Office-Home dataset, we conducted a toy experiment to visualize the feature distribution of ProDe using the t-SNE tool.

Meanwhile, five comparisons are considered, including CLIP-V, SHOT, TPDS, DIFO-V and Oracle. Among them, CLIP-V is the zero-shot result, and Oracle is trained on target domain Ar with the ground truth . For a clear view, all results are presented in 3D density charts. As shown in Fig. 3, from CLIP-V to Oracle, category clustering becomes increasingly apparent. The distribution shape of DIFO-V and ProDe-V is closer to the expert model than that of non-multimodal methods, SHOT and TPDS. Furthermore, although DIFO-V and ProDe-V have a similar pattern, ProDe-V's shape is more detailed with Oracle.

**Ablation studies.** This part isolates the effect of (1) the objective components in Eq. (6) and (2) proxy denoising (PD). Tab. 8 presents the ablation study results, with the baseline being the results of the source model (1 row). When $\mathcal{L}_{\mathrm{Apt}}$ or $\mathcal{L}_{\mathrm{Ref}}$ is used alone (2, 3 row), their performances show similar average accuracy. However, when they work together, the best results are achieved (4 row). This comparison indicates that the proposed two losses jointly contribute to the final performance. Additionally, we further evaluate the mutual information item $\mathrm{MI}(\cdot, \cdot)$ in $L_{\mathrm{Apt}}$ with a variant of ProDe, denoted ProDe w KL, where $\mathrm{MI}(\cdot, \cdot)$ is replaced by the KL divergence loss. A significant average gap of **7.2%** (compared with the results in 4 row) confirms the advantage of the mutual information optimization (5 row).

Furthermore, removing proxy denoising from the model (ProDe-V w/o PD in 6 row) leads to a decrease in average accuracy by **1.1**%, which confirms its effectiveness. To evaluate the effect of components in the proxy denoising design, we respectively remove the source and target models' logits (see Eq. (5)) to obtain two ProDe variation methods, ProDe-V w/o PD-source and ProDe-V w/o PD-target. As listed in 7 and 8 rows, using either adjustment alone led to a significant decrease in performance. Also, we perform the correction at the probability level, instead of the logit level, in another comparison ProDe-V w/o PD-logits. The average **3.1%** decrease (compared with ProDe-V's results in 4 row) confirms the rationality of correction based on logits (9 row).

Table 4: Comparison results with CLIP (%). `Appendix E.1` presents the full results.

| Method | Office-31 | Office-Home | VisDA | DomainNet-126 |
|---|---|---|---|---|
| CLIP-R | 71.4 | 72.1 | 83.7 | 72.7 |
| **ProDe**-R | **89.9** | **81.1** | **88.7** | **81.5** |
| CLIP-V | 79.8 | 76.1 | 82.9 | 76.3 |
| **ProDe**-V | **92.6** | **84.5** | **91.0** | **85.0** |

Table 5: Partial-set and open-set results (%) on **Office-Home**. `Appendix E.1` presents the full results.

| Partial-set | Venue | Avg. | Open-set | Venue | Avg. |
|---|---|---|---|---|---|
| Source | – | 62.8 | Source | – | 46.6 |
| SHOT | ICML20 | 79.3 | SHOT | ICML20 | 72.8 |
| HCL | NIPS21 | 79.6 | HCL | NIPS21 | 72.6 |
| CoWA | ICML22 | 83.2 | CoWA | ICML22 | 73.2 |
| AaD | NIPS22 | 79.7 | AaD | NIPS22 | 71.8 |
| CRS | CVPR23 | 80.6 | CRS | CVPR23 | 73.2 |
| DIFO-V | CVPR24 | 84.1 | DIFO-V | CVPR24 | 75.9 |
| **ProDe**-V | – | **84.2** | **ProDe**-V | – | **82.6** |

Table 6: Generalized SFDA results (%) on **Office-Home**. S, T are the results of the adapted target model on the source and target domains, i.e., $Acc_s$, $Acc_t$, respectively; **WAD** means With Anti-forgetting Design. `Appendix E.1` presents the full results.

| Method | Venue | WAD | Avg. S (98.1-S) | T | H |
|---|---|---|---|---|---|
| Source | – | ✗ | 98.1 | 59.2 | 73.1 |
| SHOT | ICML20 | ✗ | 84.2 (13.9) | 71.8 | 77.5 |
| GKD | IROS21 | ✗ | 86.8 (11.3) | 72.5 | 79.0 |
| NRC | NIPS21 | ✗ | 91.3 (6.8) | 72.3 | 80.7 |
| AdaCon | CVPR22 | ✗ | 88.2 (9.9) | 65.0 | 74.8 |
| CoWA | ICML22 | ✗ | 91.8 (6.3) | 72.4 | 81.0 |
| PLUE | CVPR23 | ✗ | **96.3** (1.8) | 66.9 | 79.0 |
| TPDS | IJCV24 | ✗ | 83.8 (14.3) | 73.5 | 78.3 |
| GDA | ICCV21 | ✓ | 80.0 (18.1) | 70.2 | 74.4 |
| PSAT-ViT | TMM24 | ✓ | 86.4 (11.7) | 83.6 | **85.0** |
| DIFO-R | CVPR24 | ✗ | 78.3 (19.8) | 79.4 | 78.8 |
| DIFO-V | CVPR24 | ✗ | 78.0 (20.1) | 83.1 | 80.5 |
| **ProDe**-R | – | ✗ | 84.9 (13.2) | 81.1 | 82.9 |
| **ProDe**-V | – | ✗ | 85.1 (13.0) | **84.5** | 84.8 |

Table 7: SF-MTDA, SF-MSDA and TTA results (%) on **Office-Home**. The full results of TTA are provided in `Appendix E.1`.

| | Model | Venue | Ar→ | Cl→ | Pr→ | Rw→ | Avg. | | Method | Venue | Avg. |
|---|---|---|---|---|---|---|---|---|---|---|---|
| **SF-MTDA** | CoNMix | WACV23 | 75.6 | 81.4 | 71.4 | 73.4 | 75.4 | | Tent | ICLR20 | 61.7 |
| | **ProDe**-V | – | **83.3** | **89.2** | **80.9** | **81.2** | **83.6** | | T3A | NeurIPS21 | 63.8 |
| | Method | Venue | →Rw | →Pr | →Cl | →Ar | Avg. | **TTA** | CoTTA | CVPR22 | 60.5 |
| **SF-MSDA** | SHOT-Ens | ICML20 | 82.9 | 82.8 | 59.3 | 72.2 | 74.3 | | EATA | ICML22 | 60.7 |
| | DECISION | CVPR21 | 83.6 | 84.4 | 59.4 | 74.5 | 75.5 | | SAR | ICLR23 | 60.3 |
| | **ProDe**-V-Ens | – | **91.1** | **92.5** | **73.4** | **83.0** | **85.0** | | **ProDe**-V | – | **76.5** |

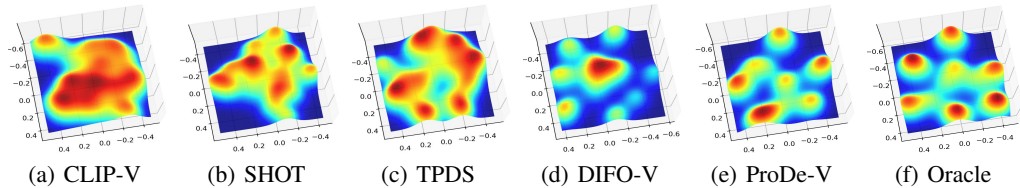

|(a) CLIP-V|(b) SHOT|(c) TPDS|(d) DIFO-V|(e) ProDe-V|(f) Oracle|

Figure 3: Feature visualization comparison in 3D density charts.

Table 8: Ablation study results (%) on **Office-31**, **Office-Home** and **VisDA**.

| # | $L_{\mathrm{Apt}}$ | $L_{\mathrm{Ref}}$ | Office-31 | Office-Home | VisDA | Avg. |
|---|---|---|---|---|---|---|
| 1 | ✗ | ✗ | 78.6 | 59.2 | 49.2 | 62.3 |
| 2 | ✓ | ✗ | 91.3 | 77.5 | 90.7 | 86.5 |
| 3 | ✗ | ✓ | 87.3 | 80.5 | 87.3 | 85.0 |
| 4 | ✓ | ✓ | **92.6** | **84.5** | **91.0** | **89.3** |
| 5 | **ProDe**-V w KL | | 83.7 | 72.9 | 89.8 | 82.1 |
| 6 | **ProDe**-V w/o PD | | 91.6 | 82.3 | 90.6 | 88.2 |
| 7 | **ProDe**-V w/o PD-source | | 91.5 | 83.5 | 90.9 | 88.6 |
| 8 | **ProDe**-V w/o PD-target | | 91.0 | 82.3 | 89.9 | 87.7 |
| 9 | **ProDe**-V w/o PD-logits | | 88.6 | 81.5 | 88.5 | 86.2 |

Table 9: Comparison results (%) on **Office-31**, **Office-Home** and **VisDA** as image encoder backbone in CLIP adopts architecture ViT-B/16. **SF** means source-free.

| Method | Venue | SF | Office-31 | Office-Home | VisDA |
|---|---|---|---|---|---|
| CLIP-V16 | ICML21 | ✗ | 77.6 | 80.1 | 85.6 |
| DAPL-V16 | TNNLS23 | ✗ | – | 85.8 | 89.8 |
| ADCLIP-V16 | ICCVW23 | ✗ | – | 86.1 | 90.7 |
| PAD-V16 | AAAI24 | ✗ | 91.2 | 85.7 | 89.7 |
| DAMP-V16 | CVPR24 | ✗ | – | 87.1 | 90.9 |
| DIFO-V16 | CVPR24 | ✓ | **92.2** | 85.5 | 91.0 |
| **ProDe-V16** | – | ✓ | **92.2** | **86.9** | **91.7** |

**Impact of image encoder backbone in CLIP.** In addition to the ResNet and ViT-B/32 architectures aforementioned, we also implement ProDe using another well-known architecture, ViT-B/16, which we refer to as ProDe-V16. Furthermore, we compare the performance of CLIP-V16, DAPL-V16, ADCLIP-V16, PAD-V16, DAMP-V16 and DIFO-V16, which also use ViT-B/16 as their image encoder. As listed in Tab. 9, ProDe-V16 still surpasses all comparisons. Combining with the ResNet and ViT-B/32 results reported in Tab. 1~Tab. 2, it is concluded that the advantage of ProDe is robust to the selection of the image-encoder backbone.

**Comparison with SFDA methods with ViT backbone.** To achieve a comprehensive evaluation, in this part, we present comparisons with typical SFDA methods using ViT backbones (cited from DPC (Zhan et al., 2024)), employing ViT-B/16. Specifically, the comparison methods include SHOT-ViT (Liang et al., 2020), DIPE-ViT (Wang et al., 2022a), DSiT-ViT (Sanyal et al., 2023), AaD-ViT (Yang et al., 2022) and DPC. The results in Tab. 10 show that ProDe-V16 consistently outperforms DPC in most cases. An exception is that ProDe-V16

Table 10: Comparison with SFDA methods with ViT backbone on closed-set SFDA setting (%).

| Method | Venue | Office-31 | Office-Home | VisDA | DomainNet-126 |
|---|---|---|---|---|---|
| SHOT-ViT | ICML20 | 91.4 | 78.1 | – | 71.4 |
| DIPE-ViT | CVPR22 | 90.5 | 78.2 | – | – |
| DSiT-ViT | ICCV23 | 93.0 | 80.5 | – | – |
| AaD-ViT | NeurIPS22 | – | – | – | 72.7 |
| DPC | IJCAI24 | **93.3** | 85.4 | – | 85.6 |
| **ProDe-V16** | – | 92.2 | **86.9** | **91.7** | **88.1** |

is only **1.1**% behind on Office-31, which may be attributed to potential overfitting on this relatively small dataset. Notably, even with a ResNet backbone for the target model, ProDe-V16 still surpasses DPC, which utilizes a ViT. Generally, using a ViT for such a small training dataset is unnecessary due to the tendency for overfitting.

## 4.4 QUANTITATIVE ANALYSIS OF PROXY DENOISING IN PROXY ALIGNMENT VIEW

In this part, we make a feature space shift analysis using the measure of MMD (Maximum Mean Discrepancy) distance to verify whether our ProDe method ensures the proxy alignment. In this experiment, we initially train a domain-invariant Oracle model over all Office-Home data with real labels, and use the logits to express the domain-invariant space $\mathcal{O}$. Sequentially, we perform a transfer experiment of Ar→Cl. During this adaptation, there are $K$ (epoch number) intermediate adapting target models. We feedforward the target data through each intermediate model and take the logits as a space. Thus, we obtain $K$ intermediate target feature spaces $\{\mathcal{U}_k\}_{k=1}^{K}$. These intermediate spaces can lead to three different kinds of distances corresponding to these frozen spaces, termed $\boldsymbol{d}_{\mathcal{S}}^t$ (to the source domain), $\boldsymbol{d}_{\mathcal{O}}^t$ (to the Oracle space) and $\boldsymbol{d}_{\mathcal{V}}^t$ (to the proxy CLIP space). In practice, the CLIP image encoder's backbone is set to ViT-B/32.

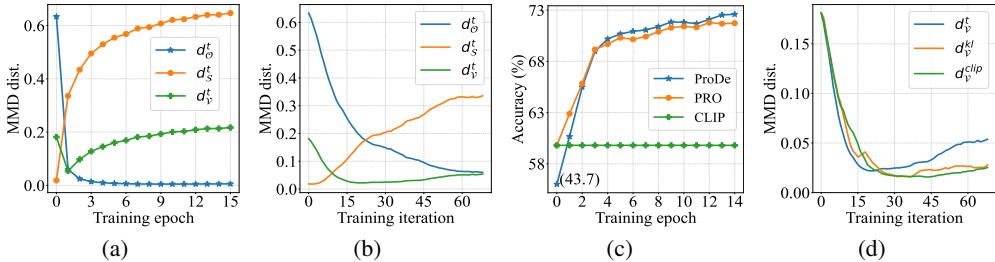

Figure 4: Analysis for proxy denoising on the Al→Cl task in Office-Home. (a) The MMD-distance varying curves (epoch view) between the intermediate spaces to the source, oracle and proxy CLIP spaces, respectively, i.e., $\boldsymbol{d}_{\mathcal{S}}^t$, $\boldsymbol{d}_{\mathcal{O}}^t$ and $\boldsymbol{d}_{\mathcal{V}}^t$. (b) The details of $\boldsymbol{d}_{\mathcal{V}}^t$ (iteration view) during the first epoch. (c) The accuracy curves of typical signals during the adaptation. (d) The MMD-distance varying curves of ProDe ($\boldsymbol{d}_{\mathcal{V}}^t$), ProDe-KL ($\boldsymbol{d}_{\mathcal{V}}^{kl}$), ProDe-CLIP ($\boldsymbol{d}_{\mathcal{V}}^{clip}$) during the first epoch (iteration view).

Fig. 4 (a) displays the varying curves (epoch view) of $\boldsymbol{d}_{\mathcal{S}}^t$, $\boldsymbol{d}_{\mathcal{O}}^t$ and $\boldsymbol{d}_{\mathcal{V}}^t$. As expected, $\boldsymbol{d}_{\mathcal{S}}^t$ increases, along with a decreasing on $\boldsymbol{d}_{\mathcal{O}}^t$. Meanwhile, $\boldsymbol{d}_{\mathcal{V}}^t$ exhibits a V-shaped trend. For a clear view, we zoom into the first epoch and observe its variation details, as shown in Fig. 4 (b). In particular, there is a smooth transition from decrease to increase on the curve of $\boldsymbol{d}_{\mathcal{V}}^t$. This phenomenon indicates that the in-training model indeed approaches the proxy space and then moves away from it to close the domain-invariant space as our proxy error control gradually comes into play.

Correspondingly, we also provide the accuracy varying curves of two typical signals in Fig. 4 (c), including the target prediction (termed ProDe) and the denoised CLIP prediction (termed PRO). In this experiment, CLIP zero-shot result (termed CLIP) is the baseline. It is seen that PRO is better than ProDe in the early phase (0∼4 epoch) and surpassed by ProDe in the rest epochs. The results indicate that the guidance of reliable ViL predictions can boost the adaptation performance. Meanwhile, the PRO and ProDe curves closely resemble each other. It is understandable that the current prediction of the in-training target model, $\theta_t(\boldsymbol{x}_i)$, is utilized to adjust the raw ViL prediction (see Eq. (5)).

To better understand the impact of proxy denoising, we also conduct a comparison using two variations of ProDe. In ProDe-KL, the loss $L_{Apt}$ is changed to conventional KL-Divergence, whilst in ProDe-CLIP, the training is based on the raw ViL prediction without proxy denoising. Employing the same MMD-distance quantification method mentioned above, we can plot two distance curves to the proxy space, termed $\boldsymbol{d}_{\mathcal{V}}^{kl}$, $\boldsymbol{d}_{\mathcal{V}}^{clip}$. In Fig. 4 (d), it is evident that ProDe moves away from the proxy space more quickly than the other two comparisons. This result suggests that ProDe is more responsive to proxy errors, resulting in agile error correction to match desired adapting direction. Additionally, the three curves at the early iterations are similar, indicating the impact of denoising $\boldsymbol{e}_{\mathcal{VI}}$ is negligible during this stage. This observation provides empirical evidence supporting Case1 in our assumption.

## 5 CONCLUSION

The success of multimodal foundation models has sparked interest in transferring general multimodal knowledge to assist with domain-specific tasks, particularly in the field of transfer learning. However, for label-free scene scenarios such as SFDA discussed in this paper, the issue of filtering out noise from multimodal foundation models has been largely overlooked. To address this fundamental issue, this paper introduces a new ProDe approach. We first introduce a new approach called proxy denoising, which corrects the raw ViL predictions and provides reliable ViL guidance. This approach is based on a novel proxy confidence theory that we developed by modeling the impact of the proxy error between the proxy ViL space and the latent domain-invariant space, using the adaptation dynamics in the proxy alignment. Additionally, we propose a mutual distilling method to make use of the reliable proxy. Extensive experiment results indicate that our ProDe can achieve state-of-the-art results with significant improvements on four challenging datasets, confirming its effectiveness.

## ACKNOWLEDGMENTS

This work is supported by the National Natural Science Foundation of China (62476169, 62206168, 62276048) and the Postdoctoral Fellowship Program of CPSF (GZC20233323).

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

## A  PROOF OF THEOREM 1

**Restatement of Theorem 1** *Given a proxy alignment formulated in* `Sec.3.1`*. The source domain* $(D_\mathcal{S})$*, the domain-invariant space* $(D_\mathcal{I})$*, the proxy space* $(D_\mathcal{V})$ *and the in-training model* $(D_\mathcal{T}^t)$ *satisfy the probability distributions* $P(\mathcal{S})$*,* $P(\mathcal{I})$*,* $P(\mathcal{V})$ *and* $P(\mathcal{T}^t)$*, respectively, where* $\mathcal{S}, \mathcal{I}, \mathcal{V}$ *and* $\mathcal{T}^t$ *are corresponding random variables. The factor describing the credibility of* $P(\mathcal{V})$ *has a relation below.*

$$P\left(G_{P(\mathcal{V})} = True, t\right) \propto \frac{P(\mathcal{T}^t)}{P(\mathcal{S})}.$$

**Proof 1** *We use the spatial distance relation to represent the variation in confidence of ViL prediction, which is causally linked to the variation in distance to* $D_\mathcal{I}$*, as demonstrated in Fig. 2 (a). At any given time* $t$*, the correction factor can be expressed as*

$$P\left(G_{P(\mathcal{V})} = True, t\right) \propto \frac{|Distance(D_\mathcal{T}^t, D_\mathcal{I})|}{|Distance(D_\mathcal{S}, D_\mathcal{I})|} = \frac{|\boldsymbol{d}_\mathcal{I}^t|}{|\boldsymbol{d}_\mathcal{S}|}. \tag{7}$$

*where* $\boldsymbol{d}_\mathcal{I}^t$ *and* $\boldsymbol{d}_\mathcal{S}$ *refers to the distance from* $D_\mathcal{T}^t$ *and* $D_\mathcal{S}$ *to* $D_\mathcal{I}$*, respectively. Easily finding, Eq. (7) satisfies the reliability feature of gradually decreasing from 1 to 0 as* $D_\mathcal{T}^t$ *evolves from* $D_\mathcal{S}$ *to* $D_\mathcal{I}$*.*

*To account for the fact that spaces are defined by probability distributions, we instantiate the space distance using the widely used measurement of KL-divergence. This gives us:*

$$\begin{aligned}
\frac{|\boldsymbol{d}_\mathcal{I}^t|}{|\boldsymbol{d}_\mathcal{S}|} &= \frac{KL\left(P(\mathcal{T}^t)||P(\mathcal{I})\right)}{KL\left(P(\mathcal{S})||P(\mathcal{I})\right)} = \frac{\int_{\mathcal{T}^t} P(\mathcal{T}^t) \log \frac{P(\mathcal{T}^t)}{P(\mathcal{I})} d\mathcal{T}^t}{\int_{\mathcal{S}} P(\mathcal{S}) \log \frac{P(\mathcal{S})}{P(\mathcal{I})} d\mathcal{S}} \\
&= \frac{-\int_{\mathcal{T}^t} P(\mathcal{T}^t) \log P(\mathcal{T}^t) d\mathcal{T}^t + \int_{\mathcal{T}^t} P(\mathcal{T}^t) \log P(\mathcal{I}) d\mathcal{T}^t}{-\int_{\mathcal{S}} P(\mathcal{S}) \log P(\mathcal{S}) d\mathcal{S} + \int_{\mathcal{S}} P(\mathcal{S}) \log P(\mathcal{I}) d\mathcal{S}} \\
&= \frac{H(\mathcal{T}^t) + \log P(\mathcal{I})}{H(\mathcal{S}) + \log P(\mathcal{I})}
\end{aligned} \tag{8}$$

*where* $H(\cdot)$ *stands for the information entropy. Since* $D_\mathcal{I}$ *is an domain-invariant space,* $P(\mathcal{I})$ *always outputs 1 for the category of interesting, such that* $\log P(\mathcal{I}) = 0$*. Eq. (8) can be further converted to*

$$\frac{H(\mathcal{T}^t) + \log P(\mathcal{I})}{H(\mathcal{S}) + \log P(\mathcal{I})} = \frac{H(\mathcal{T}^t)}{H(\mathcal{S})} \propto \frac{P(\mathcal{T}^t)}{P(\mathcal{S})} \tag{9}$$

## B  PSEUDO TRAINING CODE OF PRODE

Based on the proposed objective presented in Eq. (6), we achieve the model training iteration-wise. The training process are summarized as Alg. 1.

---
**Algorithm 1** Training of ProDe
---
**Input**: Source model $\theta_s$, ViL model $\theta_v$, target dataset $\mathcal{X}_\mathcal{T}$, $C$ prompts with context $\boldsymbol{v}$, #iteration $M$.
**Procedure**:
 1: **Initialisation**: Set target model $\theta_t = \theta_s$, prompt context $\boldsymbol{v} =$"a photo of a".
 2: **for** $m$ = 1:$M$ **do**
 3:     Sample a batch $\mathcal{X}_\mathcal{T}^b$ from $\mathcal{X}_\mathcal{T}$.
 4:     Forward updated prompts and $\mathcal{X}_\mathcal{T}^b$ through $\theta_v$.
 5:     Forward $\mathcal{X}_\mathcal{T}^b$ through $\theta_t$.
 6:     Conduct proxy denoising for the ViL predictions of $\mathcal{X}_\mathcal{T}^b$ (Eq. (5)).
 7:     Update model $\theta_t$ and prompt context $\boldsymbol{v}$ by optimizing objective $L_{\mathrm{ProDe}}$ (Eq. (6)).
 8: **end for**
 9: **return** Adapted target model $\theta_t$.
---

## C  EVALUATION DATASETS

In this paper, the ProDe method is evaluated on four widely used benchmarks for domain adaptation problems as follows.

- **Office-31** (Saenko et al., 2010) is a small-scaled dataset including three domains, i.e., Amazon (A), Webcam (W), and Dslr (D), all of which are taken of real-world objects in various office environments. The dataset has 4,652 images of 31 categories in total.

- **Office-Home** (Venkateswara et al., 2017) is a medium-scale dataset that is mainly used for domain adaptation, all of which contains 15k images belonging to 65 categories from working or family environments. The dataset has four distinct domains, i.e., Artistic images (Ar), Clip Art (Cl), Product images (Pr), and Real-word images (Rw).

- **VisDA** (Peng et al., 2017) is a large-scale dataset with 12 types of synthetic to real transfer recognition tasks. The source domain contains 152k synthetic images (Sy), whilst the target domain has 55k real object images (Re) from the famous Microsoft COCO dataset.

- **DomainNet-126** (Saito et al., 2019) is another challenging large-scale dataset. It has been created by removing severe noisy labels from the original DomainNet dataset (Peng et al., 2019) containing 600k images of 345 classes from 6 domains of varying image styles. The dataset is further divided into four domains: Clipart (C), Painting (P), Real (R), and Sketch (S), and contains 145k images from 126 classes.

## D    IMPLEMENTATION DETAILS

**Souce model pre-training.** For all transfer tasks on the four evaluation datasets, we train the source model $\theta_s$ on the source domain in a supervised manner using the following objective of the classic cross-entropy loss with smooth label, totally the same as other methods (Liang et al., 2020; Yang et al., 2021a; Tang et al., 2022).

$$L_s \left( \mathcal{X}_s, \mathcal{Y}_s; \theta_s \right) = -\mathbb{E}_{\boldsymbol{x}_i^s \in \mathcal{X}_s} \sum_{c=1}^{C} \tilde{\mathbb{1}} \left[ c = y_i^s \right] \log p_{i,c}^s,$$

where $p_{i,c}^s$ is the $c$-th element of $\boldsymbol{p}_i^s = \phi(\theta_s(\boldsymbol{x}_i^s))$ that is the category probability vector of input instance $\boldsymbol{x}_i^s$ after $\theta_s$ conversion with ending softmax operation $\phi$; $\tilde{\mathbb{1}} \left[ c = y_i^s \right] = (1 - \sigma) \mathbb{1} \left[ c = y_i^s \right] + \sigma/C$ is the smooth label (Müller et al., 2019), in which $\mathbb{1} \left[ c = y_i^s \right]$ is a one-hot encoding of hard label $y_i^s$ and $\sigma = 0.1$. The source dataset is divided into the training set and testing set in a 0.9:0.1 ratio.

**Network setting.** The ProDe framework involves two networks, namely the target model and the ViL model. In practice, the target model comprises a deep architecture-based feature extractor and a classifier that consists of a fully connected layer and a weight normalization layer. As seen in previous work (Xu et al., 2019; Liang et al., 2020; Roy et al., 2022), the deep architecture is transferred from the deep models pre-trained on ImageNet. Specifically, ResNet-50 is used on Office-31 and Office-Home, whilst ResNet-101 is employed on VisDA and Domain-Net. As for the ViL model, we choose CLIP to instantiate it where the text encoder adopts Transformer structure and the image encoder takes ResNet or ViT-B/32 according to the specific implementations, which are marked by suffix of "-R" or "-V".

**Hyper-parameter setting.** The ProDe model involves four parameters: The correction strength factor $\omega$ in Eq. (5) and two trade-off parameters $\alpha$, $\beta$ and $\gamma$ in objective $L_{ProDe}$ (Eq. (6)). On all four datasets, we set $(\omega, \alpha, \beta) = (1, 1, 0.4)$. Parameter $\gamma$ is sensitive to the dataset scale, also noted in the TPDS method (Tang et al., 2024a). In practice, the setting of $\gamma = 1.0/1.0/0.1/0.5$ is employed on Office-31, Office-Home, VisDA and DomainNet-126, respectively.

**Training setting.** We chose a batch size of 64 and utilized the SGD optimizer with a momentum of 0.9 and 15 training epochs on all datasets. The learnable prompt context is initiated by the template of 'a photo of a [CLASS].', as suggested by (Radford et al., 2021), where the [CLASS] term is replaced with the name of the class being trained. All experiments are conducted with PyTorch on a single GPU of NVIDIA RTX. Each transfer task is repeated five times, and the final result is calculated as the average of the five attempts.

Table 11: Full results (%) of closed-set SFDA on **VisDA**. **SF** means source-free.

| Method | Venue | SF | plane | bcycl | bus | car | horse | knife | mcycl | person | plant | sktbrd | train | truck | Perclass |
|---|---|---|---|---|---|---|---|---|---|---|---|---|---|---|---|
| Source | - | - | 60.7 | 21.7 | 50.8 | 68.5 | 71.8 | 5.4 | 86.4 | 20.2 | 67.1 | 43.3 | 83.3 | 10.6 | 49.2 |
| SHOT | ICML20 | ✓ | 95.0 | 87.4 | 80.9 | 57.6 | 93.9 | 94.1 | 79.4 | 80.4 | 90.9 | 89.8 | 85.8 | 57.5 | 82.7 |
| NRC | NIPS21 | ✓ | 96.8 | 91.3 | 82.4 | 62.4 | 96.2 | 95.9 | 86.1 | **90.7** | 94.8 | 94.1 | 90.4 | 59.7 | 85.9 |
| GKD | IROS21 | ✓ | 95.3 | 87.6 | 81.7 | 58.1 | 93.9 | 94.0 | 80.0 | 80.0 | 91.2 | 91.0 | 86.9 | 56.1 | 83.0 |
| AaD | NIPS22 | ✓ | 97.4 | 90.5 | 80.8 | 76.2 | 97.3 | 96.1 | 89.8 | 82.9 | 95.5 | 93.0 | 92.0 | 64.7 | 88.0 |
| AdaCon | CVPR22 | ✓ | 97.0 | 84.7 | 84.0 | 77.3 | 96.7 | 93.8 | 91.9 | 84.8 | 94.3 | 93.1 | 94.1 | 49.7 | 86.8 |
| CoWA | ICML22 | ✓ | 96.2 | 89.7 | 83.9 | 73.8 | 96.4 | 97.4 | 89.3 | 86.8 | 94.6 | 92.1 | 88.7 | 53.8 | 86.9 |
| ELR | ICLR23 | ✓ | 97.1 | 89.7 | 82.7 | 62.0 | 96.2 | 97.0 | 87.6 | 81.2 | 93.7 | 94.1 | 90.2 | 58.6 | 85.8 |
| PLUE | CVPR23 | ✓ | 94.4 | 91.7 | 89.0 | 70.5 | 96.6 | 94.9 | 92.2 | 88.8 | 92.9 | 95.3 | 91.4 | 61.6 | 88.3 |
| CPD | PR24 | ✓ | 96.7 | 88.5 | 79.6 | 69.0 | 95.9 | 96.3 | 87.3 | 83.3 | 94.4 | 92.9 | 87.0 | 58.7 | 85.5 |
| TPDS | IJCV24 | ✓ | 97.6 | 91.5 | 89.7 | 83.4 | 97.5 | 96.3 | 92.2 | 82.4 | **96.0** | 94.1 | 90.9 | 40.4 | 87.6 |
| DAPL-R | TNNLS23 | ✗ | 97.8 | 83.1 | 88.8 | 77.9 | 97.4 | 91.5 | 94.2 | 79.7 | 88.6 | 89.3 | 92.5 | 62.0 | 86.9 |
| PADCLIP-R | ICCV23 | ✗ | 96.7 | 88.8 | 87.0 | 82.8 | 97.1 | 93.0 | 91.3 | 83.0 | 95.5 | 91.8 | 91.5 | 63.0 | 88.5 |
| ADCLIP-R | ICCVW23 | ✗ | 98.1 | 83.6 | **91.2** | 76.6 | 98.1 | 93.4 | **96.0** | 81.4 | 86.4 | 91.5 | 92.1 | 64.2 | 87.7 |
| PDA-R | AAAI24 | ✗ | 97.2 | 82.3 | 89.4 | 76.0 | 97.4 | 87.5 | 95.8 | 79.6 | 87.2 | 89.0 | 93.3 | 62.1 | 86.4 |
| DAMP-R | CVPR24 | ✗ | 97.3 | 91.6 | 89.1 | 76.4 | 97.5 | 94.0 | 92.3 | 84.5 | 91.2 | 88.1 | 91.2 | 67.0 | 88.4 |
| DIFO-R | CVPR24 | ✓ | 97.6 | 88.7 | 83.7 | 80.8 | 95.9 | 95.3 | 91.9 | 85.0 | 89.4 | 93.2 | 93.2 | 69.0 | 88.6 |
| DIFO-V | CVPR24 | ✓ | 97.5 | 89.0 | 90.8 | **83.5** | 97.8 | 97.3 | 93.2 | 83.5 | 95.2 | 96.8 | 93.7 | 65.9 | 90.3 |
| **ProDe-R** | – | ✓ | 96.6 | 90.3 | 83.9 | 80.2 | 96.1 | 96.9 | 90.3 | 86.4 | 90.8 | 94.0 | 91.3 | 67.0 | 88.7 |
| **ProDe-V** | – | ✓ | **98.3** | **92.4** | 86.6 | 80.5 | **98.1** | **98.0** | 92.3 | 84.3 | 94.7 | **97.0** | **94.1** | **75.6** | **91.0** |

Table 12: Full results (%) of partial-set SFDA and open-set SFDA on **Office-Home**.

| Partial-set | Venue | Ar→Cl | Ar→Pr | Ar→Rw | Cl→Ar | Cl→Pr | Cl→Rw | Pr→Ar | Pr→Cl | Pr→Rw | Rw→Ar | Rw→Cl | Rw→Pr | Avg. |
|---|---|---|---|---|---|---|---|---|---|---|---|---|---|---|
| Source | – | 45.2 | 70.4 | 81.0 | 56.2 | 60.8 | 66.2 | 60.9 | 40.1 | 76.2 | 70.8 | 48.5 | 77.3 | 62.8 |
| SHOT | ICML20 | 64.8 | 85.2 | 92.7 | 76.3 | 77.6 | 88.8 | 79.7 | 64.3 | 89.5 | 80.6 | 66.4 | 85.8 | 79.3 |
| HCL | NIPS21 | 65.6 | 85.2 | 92.7 | 77.3 | 76.2 | 87.2 | 78.2 | 66.0 | 89.1 | 81.5 | 68.4 | 87.3 | 79.6 |
| CoWA | ICML22 | 69.6 | **93.2** | 92.3 | 78.9 | 81.3 | **92.1** | 79.8 | **71.7** | 90.0 | 83.8 | **72.2** | **93.7** | 83.2 |
| AaD | NIPS22 | 67.0 | 83.5 | **93.1** | 80.5 | 76.0 | 87.6 | 78.1 | 65.6 | 90.2 | 83.5 | 64.3 | 87.3 | 79.7 |
| CRS | CVPR23 | 68.6 | 85.1 | 90.9 | 80.1 | 79.4 | 86.3 | 79.2 | 66.1 | 90.5 | 82.2 | 69.5 | 89.3 | 80.6 |
| DIFO-V | CVPR24 | 69.9 | 88.8 | 90.3 | **85.7** | 89.5 | 91.2 | **85.8** | 70.3 | 92.8 | 87.1 | 69.1 | 89.1 | 84.1 |
| **ProDe-V** | – | **70.2** | 89.7 | 90.4 | 84.1 | **90.7** | 91.4 | 85.5 | 69.9 | **92.9** | **87.8** | 68.5 | 89.7 | **84.2** |
| Open-set | Venue | Ar→Cl | Ar→Pr | Ar→Rw | Cl→Ar | Cl→Pr | Cl→Rw | Pr→Ar | Pr→Cl | Pr→Rw | Rw→Ar | Rw→Cl | Rw→Pr | Avg. |
| Source | – | 36.3 | 54.8 | 69.1 | 33.8 | 44.4 | 49.2 | 36.8 | 29.2 | 56.8 | 51.4 | 35.1 | 62.3 | 46.6 |
| SHOT | ICML20 | 64.5 | 80.4 | 84.7 | 63.1 | 75.4 | 81.2 | 65.3 | 59.3 | 83.3 | 69.6 | 64.6 | 82.3 | 72.8 |
| HCL | NIPS21 | 64.0 | 78.6 | 82.4 | 64.5 | 73.1 | 80.1 | 64.8 | 59.8 | 75.3 | 78.1 | 69.3 | 81.5 | 72.6 |
| CoWA | ICML22 | 63.3 | 79.2 | 85.4 | 67.6 | 83.6 | 82.0 | 66.9 | 56.9 | 81.1 | 68.5 | 57.9 | 85.9 | 73.2 |
| AaD | NIPS22 | 63.7 | 77.3 | 80.4 | 66.0 | 72.6 | 77.6 | 69.1 | 62.5 | 79.8 | 71.8 | 62.3 | 78.6 | 71.8 |
| CRS | CVPR23 | 65.2 | 76.6 | 80.2 | 66.2 | 75.3 | 77.8 | 70.4 | 61.8 | 79.3 | 71.1 | 61.1 | 78.3 | 73.2 |
| DIFO-V | CVPR24 | 64.5 | **86.2** | **87.9** | 68.2 | 79.3 | 86.1 | 67.2 | 62.1 | **88.3** | 71.9 | 65.3 | 84.4 | 75.9 |
| **ProDe-V** | – | **75.9** | 85.6 | **87.9** | **81.3** | **86.8** | **87.2** | **81.1** | **74.3** | 86.3 | **83.0** | **75.7** | **86.1** | **82.6** |

Table 13: Results (%) of CLIP on the four evaluation datasets. The backbone of CLIP image-encoder in CLIP-R and CLIP-V are the same as ProDe-R and ProDe-V, respectively.

| Method | Venue | Office-31 | | | | Office-Home | | | | | VisDA | DomainNet-126 | | | | |
|---|---|---|---|---|---|---|---|---|---|---|---|---|---|---|---|---|
| | | →A | →D | →W | →Avg. | →Ar | →Cl | →Pr | →Rw | →Avg. | Sy→Re | →C | →P | →R | →S | →Avg. |
| CLIP-R | ICML21 | 73.1 | 73.9 | 67.0 | 71.4 | 72.5 | 51.9 | 81.5 | 82.5 | 72.1 | 83.7 | 67.9 | 70.2 | 87.1 | 65.4 | 72.7 |
| **ProDe-R** | – | **79.4** | **96.5** | **93.9** | **89.9** | **80.6** | **65.0** | **90.1** | **88.6** | **81.1** | **88.7** | **80.3** | **79.2** | **91.0** | **75.4** | **81.5** |
| CLIP-V | ICML21 | 76.0 | 82.7 | 80.6 | 79.8 | 74.6 | 59.8 | 84.3 | 85.5 | 76.1 | 82.9 | 74.7 | 73.5 | 85.7 | 71.2 | 76.3 |
| **ProDe-V** | – | **82.8** | **98.3** | **96.7** | **92.6** | **82.7** | **72.6** | **92.0** | **90.7** | **84.5** | **91.0** | **85.3** | **83.2** | **92.4** | **79.1** | **85.0** |

# E  SUPPLEMENTAL EXPERIMENTS

## E.1  SUPPLEMENTATION OF FULL EXPERIMENT RESULTS

**Full results on VisDA.** Tab. 11 is the supplement of average results on the VisDA dataet (reported in Tab. 2), displaying the full classification results over the 12 categories. Specifically, the ProDe-R and ProDe-V totally obtain best results on 7/12 categories, leading to the advantage on average accuracy. On some cases, such as bcycl, car and truck, ProDe has presents significant advantages over the previous methods.

Table 14: Generalized SFDA results (%) on **Office-Home**. S, T are the results of the adapted target domain on the source and target domains, respectively; H means the harmonic mean accuracy; **WAD** is short for With Anti-forgetting Design.

| Method | Venue | WAD | Ar→Cl S | T | H | Ar→Pr S | T | H | Ar→Rw S | T | H | Cl→Ar S | T | H | Cl→Pr S | T | H | Cl→Rw S | T | H |
|---|---|---|---|---|---|---|---|---|---|---|---|---|---|---|---|---|---|---|---|---|
| Source | – | ✗ | 97.9 | 43.7 | 60.4 | 97.9 | 67.0 | 79.5 | 97.9 | 73.9 | 84.2 | 97.1 | 49.9 | 65.9 | 97.1 | 60.1 | 74.2 | 97.1 | 62.5 | 76.0 |
| SHOT | ICML20 | ✗ | 78.6 | 55.0 | 64.7 | 83.8 | 78.7 | 81.2 | 88.6 | 81.3 | 84.8 | 78.0 | 69.1 | 73.2 | 76.6 | 78.9 | 77.7 | 77.1 | 79.1 | 78.1 |
| GKD | IROS21 | ✗ | 81.9 | 56.5 | 66.9 | 87.0 | 78.3 | 82.4 | 91.4 | 82.2 | 86.6 | 80.3 | 69.2 | 74.3 | 80.9 | 80.4 | 80.6 | 81.4 | 78.7 | 80.1 |
| NRC | NIPS21 | ✗ | 86.9 | 57.2 | 69.0 | 92.9 | 79.3 | 85.6 | 95.3 | 81.3 | 87.7 | 81.7 | 68.9 | 74.8 | 89.1 | 80.6 | 84.6 | 88.8 | 80.2 | 84.3 |
| AdaCon | CVPR22 | ✗ | 75.2 | 47.2 | 57.9 | 91.0 | 75.1 | 82.3 | 93.9 | 75.5 | 83.7 | 79.4 | 60.7 | 68.8 | 88.2 | 73.3 | 80.0 | 83.4 | 73.2 | 78.0 |
| CoWA | ICML22 | ✗ | 89.0 | 57.3 | 69.7 | 93.0 | 79.3 | 85.6 | 94.6 | 81.0 | 87.3 | 86.6 | 69.3 | 77.0 | 86.3 | 77.9 | 81.9 | 83.4 | 79.6 | 81.5 |
| PLUE | CVPR23 | ✗ | 91.8 | 49.1 | 63.9 | 96.3 | 73.5 | 83.4 | 97.2 | 78.2 | 86.6 | 93.9 | 63.0 | 75.3 | 95.6 | 73.5 | 83.1 | 94.3 | 74.5 | 83.2 |
| TPDS | IJCV24 | ✗ | 78.0 | 59.3 | 67.4 | 83.6 | 80.3 | 81.9 | 88.1 | 82.1 | 85.0 | 75.4 | 70.6 | 72.9 | 77.3 | 79.4 | 78.3 | 76.2 | 80.9 | 78.5 |
| GDA | ICCV21 | ✓ | 68.8 | 54.7 | 60.9 | 72.0 | 75.6 | 73.8 | 74.5 | 78.5 | 76.4 | 77.2 | 66.6 | 71.5 | 79.7 | 74.0 | 76.7 | 78.5 | 78.4 | 78.4 |
| PSAT-ViT | TMM24 | ✓ | 81.6 | 73.1 | 77.1 | 87.0 | 88.1 | 87.6 | 88.1 | 89.2 | 88.7 | 82.7 | 82.1 | 82.6 | 82.7 | 88.8 | 85.7 | 83.5 | 88.9 | 86.1 |
| DIFO-R | CVPR24 | ✗ | 73.8 | 62.6 | 67.8 | 76.3 | 87.5 | 81.5 | 79.7 | 87.1 | 83.2 | 73.1 | 79.5 | 76.2 | 64.8 | 87.9 | 74.6 | 66.3 | 87.4 | 75.4 |
| DIFO-V | CVPR24 | ✗ | 73.8 | 70.6 | 72.2 | 75.0 | 90.6 | 82.1 | 80.7 | 88.8 | 84.6 | 70.4 | 82.5 | 75.9 | 64.3 | 90.6 | 75.2 | 65.9 | 88.8 | 75.7 |
| **ProDe-R** | – | ✗ | 79.4 | 64.0 | 70.9 | 84.1 | 90.0 | 87.0 | 87.7 | 88.3 | 88.0 | 79.5 | 81.1 | 80.3 | 76.2 | 90.1 | 82.5 | 73.7 | 88.6 | 80.4 |
| **ProDe-V** | – | ✗ | 81.4 | 72.7 | 76.8 | 84.3 | 92.2 | 88.1 | 88.1 | 90.5 | 89.2 | 76.6 | 82.5 | 79.4 | 77.8 | 91.5 | 84.1 | 74.0 | 90.7 | 81.5 |

| Method | Venue | WAD | Pr→Ar S | T | H | Pr→Cl S | T | H | Pr→Rw S | T | H | Rw→Ar S | T | H | Rw→Cl S | T | H | Rw→Pr S | T | H | Avg. S | T | H |
|---|---|---|---|---|---|---|---|---|---|---|---|---|---|---|---|---|---|---|---|---|---|---|---|
| Source | – | ✗ | 99.2 | 51.7 | 68.0 | 99.2 | 40.9 | 57.9 | 99.2 | 72.6 | 83.8 | 98.1 | 64.2 | 77.6 | 98.1 | 46.3 | 62.9 | 98.1 | 78.1 | 87.0 | 98.1 | 59.2 | 73.1 |
| SHOT | ICML20 | ✗ | 88.2 | 68.2 | 76.9 | 80.7 | 53.6 | 64.4 | 90.1 | 81.6 | 85.6 | 91.7 | 73.5 | 81.6 | 84.8 | 59.4 | 69.8 | 92.2 | 83.5 | 87.6 | 84.2 | 71.8 | 77.5 |
| GKD | IROS21 | ✗ | 89.4 | 67.4 | 76.8 | 84.1 | 55.4 | 66.8 | 92.0 | 82.6 | 87.0 | 93.7 | 74.3 | 82.9 | 86.2 | 60.3 | 70.9 | 93.5 | 84.2 | 88.6 | 86.8 | 72.5 | 79.0 |
| NRC | NIPS21 | ✗ | 89.1 | 66.6 | 76.2 | 90.1 | 57.3 | 70.1 | 96.6 | 82.0 | 88.7 | 97.8 | 71.0 | 82.3 | 90.7 | 57.9 | 70.7 | 97.1 | 84.9 | 90.6 | 91.3 | 72.3 | 80.7 |
| AdaCon | CVPR22 | ✗ | 93.4 | 60.2 | 73.2 | 88.4 | 45.2 | 59.8 | 94.3 | 76.6 | 84.5 | 93.3 | 65.6 | 77.0 | 84.1 | 48.3 | 61.3 | 94.5 | 79.1 | 86.1 | 88.2 | 65.0 | 74.8 |
| CoWA | ICML22 | ✗ | 94.6 | 68.1 | 79.2 | 93.2 | 56.4 | 70.3 | 95.0 | 82.6 | 88.3 | 96.3 | 72.9 | 83.0 | 93.7 | 61.3 | 74.1 | 95.6 | 83.7 | 89.3 | 91.8 | 72.4 | 81.0 |
| PLUE | CVPR23 | ✗ | 98.7 | 62.2 | 76.3 | 98.5 | 48.3 | 64.8 | 98.9 | 78.6 | 87.6 | 98.1 | 51.8 | 67.1 | 97.8 | 51.5 | 67.7 | 96.3 | 66.9 | 79.0 | 96.3 | 64.3 | 76.8 |
| TPDS | IJCV24 | ✗ | 87.7 | 69.8 | 77.7 | 81.4 | 56.8 | 66.9 | 90.4 | 82.1 | 86.0 | 92.3 | 74.5 | 82.5 | 83.2 | 61.2 | 70.5 | 92.0 | 85.3 | 88.5 | 83.8 | 73.5 | 78.3 |
| GDA | ICCV21 | ✓ | 87.8 | 65.1 | 74.8 | 86.3 | 53.2 | 66.1 | 90.3 | 81.6 | 85.7 | 83.2 | 72.0 | 77.2 | 78.3 | 60.2 | 68.1 | 83.4 | 82.8 | 83.1 | 80.0 | 70.2 | 74.4 |
| PSAT-ViT | TMM24 | ✓ | 89.6 | 83.0 | 86.2 | 87.4 | 72.0 | 79.0 | 92.5 | 89.0 | 91.0 | 87.4 | 83.3 | 85.3 | 84.2 | 73.7 | 78.6 | 89.6 | 91.3 | 90.5 | 86.4 | 83.6 | 85.0 |
| DIFO-R | CVPR24 | ✗ | 85.6 | 78.3 | 81.8 | 76.6 | 63.4 | 69.4 | 86.0 | 88.1 | 87.0 | 89.4 | 80.0 | 84.4 | 80.7 | 63.3 | 70.9 | 87.2 | 87.7 | 87.4 | 78.3 | 79.4 | 78.8 |
| DIFO-V | CVPR24 | ✗ | 84.3 | 80.9 | 82.5 | 77.4 | 70.1 | 73.6 | 87.2 | 88.9 | 88.0 | 88.5 | 83.4 | 85.9 | 80.9 | 70.5 | 75.3 | 87.4 | 91.2 | 89.3 | 78.0 | 83.1 | 80.5 |
| **ProDe-R** | – | ✗ | 89.5 | 79.8 | 84.4 | 85.8 | 65.5 | 74.2 | 92.1 | 89.0 | 90.5 | 93.1 | 80.9 | 86.6 | 85.8 | 65.5 | 74.3 | 92.1 | 90.2 | 91.1 | 84.9 | 81.1 | 82.9 |
| **ProDe-V** | – | ✗ | 88.9 | 82.5 | 85.5 | 85.0 | 72.4 | 78.2 | 92.0 | 90.8 | 91.4 | 92.7 | 83.0 | 87.6 | 87.5 | 72.6 | 79.3 | 93.1 | 92.2 | 92.6 | 85.1 | 84.5 | 84.8 |

Table 15: Full results (%) of the TTA setting on **Office-Home**.

| Method | Venue | Ar→Cl | Ar→Pr | Ar→Rw | Cl→Ar | Cl→Pr | Cl→Rw | Pr→Ar | Pr→Cl | Pr→Rw | Rw→Ar | Rw→Cl | Rw→Pr | Avg. |
|---|---|---|---|---|---|---|---|---|---|---|---|---|---|---|
| Tent | ICLR20 | 47.6 | 63.2 | 72.3 | 57.1 | 63.7 | 65.9 | 55.9 | 46.6 | 72.7 | 67.7 | 51.8 | 77.1 | 61.7 |
| T3A | NeurIPS21 | 49.7 | 73.2 | 77.0 | 55.5 | 67.7 | 68.5 | 55.8 | 46.1 | 75.7 | 67.0 | 49.6 | 78.0 | 63.8 |
| CoTTA | CVPR22 | 44.5 | 62.5 | 72.3 | 55.4 | 63.0 | 65.3 | 54.9 | 46.0 | 76.7 | 66.0 | 49.5 | 76.7 | 60.5 |
| EATA | ICML22 | 46.4 | 62.5 | 72.2 | 55.3 | 65.8 | 65.8 | 53.8 | 43.4 | 76.4 | 66.5 | 50.5 | 76.4 | 60.7 |
| SAR | ICLR23 | 45.3 | 61.9 | 71.9 | 55.4 | 66.4 | 65.7 | 53.7 | 42.7 | 72.5 | 66.4 | 49.3 | 76.2 | 60.3 |
| **ProDe**-V | – | 62.3 | 83.2 | 83.5 | 74.9 | 83.8 | 83.3 | 73.8 | 63.7 | 84.5 | 76.6 | 63.3 | 85.1 | 76.5 |

Table 16: Reliance analysis results (%) on **Office-31** in the Closed-set SFDA setting.

| Method | Venue | A→D | A→W | D→A | D→W | W→A | W→D | Avg. |
|---|---|---|---|---|---|---|---|---|
| DIFO w/ CLIP | CVPR24 | 97.2 | 95.5 | 83.0 | 97.2 | 83.2 | 98.8 | 92.5 |
| **ProDe** w/ CLIP | – | 96.8 | 96.4 | 83.1 | 97.0 | 82.5 | 99.8 | 92.6 |
| DIFO w/ OpenCLIP | CVPR24 | 96.8 | 98.1 | 82.9 | 98.7 | 82.7 | 100. | 93.2 |
| **ProDe** w/ OpenCLIP | – | 96.1 | 96.7 | 86.5 | 97.4 | 86.8 | 98.2 | 93.6 |

**Full results of partial-set and open-set SFDA.** Tab. 12 is the supplementation of these average accuracy in Tab. 5, reporting the full classification accuracy over 12 transfer tasks in Office-Home. In the partial-set setting (the top in the table), ProDe-V beats other methods on 4/12 tasks, whilst DIFO-V, CoWA, and AaD dominate the rest of the tasks. As taking the open-set setting (the bottom in the table), ProDe-V gets the top results on 9/12 tasks. Moreover, besides the Ar→Rw, Cl→Rw, Rw→Pr task, the rest of the best eight tasks have **7.5**% increase at least, compared with the best-second methods. So, the ProDe gains substantial improvement in average performance.

**Full results of the comparison to CLIP's zero-shot.** As the supplement of average results in the comparison to CLIP (reported in Tab. 4), Tab. 13 presents the full quantitative results categorized by the target domain name. For instance, for domain A in Office-31, we averaged the adapting accuracy of other domains to A, such as D→A, W→A, notated by →A. As reported in Tab. 13, both ProDe-R and ProDe-V obtain the best results across all groups, compared to the respective CLIP version.

Table 17: Reliance analysis results (%) on **Office-Home** in the Closed-set SFDA setting.

| Method | Venue | Ar→Cl | Ar→Pr | Ar→Rw | Cl→Ar | Cl→Pr | Cl→Rw | Pr→Ar | Pr→Cl | Pr→Rw | Rw→Ar | Rw→Cl | Rw→Pr | Avg. |
|---|---|---|---|---|---|---|---|---|---|---|---|---|---|---|
| DIFO w/ CLIP | CVPR24 | 70.6 | 90.6 | 88.8 | **82.5** | 90.6 | 88.8 | 80.9 | 70.1 | 88.9 | **83.4** | 70.5 | 91.2 | 83.1 |
| **ProDe** w/ CLIP | – | **72.7** | **92.3** | **90.5** | 82.5 | **91.5** | **90.7** | **82.5** | **72.5** | **90.8** | 83.0 | **72.6** | **92.2** | **84.5** |
| DIFO w/ OpenCLIP | CVPR24 | 80.2 | 94.2 | 91.7 | 85.4 | 93.7 | 91.6 | 82.7 | 79.2 | 91.7 | 85.3 | 80.4 | 94.8 | 87.6 |
| **ProDe** w/ OpenCLIP | – | **82.3** | **95.2** | **93.1** | **87.2** | **95.5** | **93.5** | **86.9** | **82.3** | **93.5** | **87.6** | **82.6** | **95.8** | **89.6** |

Table 18: Reliance analysis results (%) on **VisDA** in the Closed-set SFDA setting.

| Method | Venue | plane | bcycl | bus | car | horse | knife | mcycl | person | plant | sktbrd | train | truck | Perclass |
|---|---|---|---|---|---|---|---|---|---|---|---|---|---|---|
| DIFO w/ CLIP | CVPR24 | 97.5 | 89.0 | **90.8** | **83.5** | 97.8 | 97.3 | **93.2** | 83.5 | **95.2** | 96.8 | 93.7 | 65.9 | 90.3 |
| **ProDe** w/ CLIP | – | **98.3** | **92.4** | 86.6 | 80.5 | **98.1** | **98.0** | 92.3 | **84.3** | 94.7 | **97.0** | **94.1** | **75.6** | **91.0** |
| DIFO w/ OpenCLIP | CVPR24 | 98.3 | 91.6 | **90.8** | 81.7 | **97.9** | **98.3** | 92.4 | 87.5 | 92.1 | 95.8 | **93.6** | 68.4 | 90.7 |
| **ProDe** w/ OpenCLIP | – | 97.9 | 90.5 | 86.7 | **88.5** | 97.8 | 96.4 | **94.2** | **88.4** | **95.7** | **96.2** | 93.3 | **71.5** | **91.4** |

Table 19: Reliance analysis results (%) on **DomainNet-126** in the Closed-set SFDA setting.

| Method | Venue | C→P | C→R | C→S | P→C | P→R | P→S | R→C | R→P | R→S | S→C | S→P | S→R | Avg. |
|---|---|---|---|---|---|---|---|---|---|---|---|---|---|---|
| DIFO w/ CLIP | CVPR24 | 76.6 | 87.2 | 74.9 | 80.0 | 87.4 | 75.6 | 80.8 | 77.3 | 75.5 | 80.5 | 76.7 | 87.3 | 80.0 |
| **ProDe** w/ CLIP | – | **83.2** | **92.4** | **79.0** | **85.0** | **92.3** | **79.3** | **85.5** | **83.1** | **79.1** | **85.5** | **83.4** | **92.4** | **85.0** |
| DIFO w/ OpenCLIP | CVPR24 | 91.2 | 91.5 | 79.4 | 85.2 | 91.2 | 79.7 | 85.7 | 82.7 | 80.5 | 85.9 | 81.3 | 91.4 | 84.6 |
| **ProDe** w/ OpenCLIP | – | 86.7 | **93.7** | **84.4** | **89.2** | **93.7** | **84.5** | **89.6** | **86.6** | **84.4** | **89.5** | **86.7** | **93.7** | **88.6** |

Table 20: Results (%) of OpenCLIP on the four evaluation datasets.

| Method | Venue | Office-31 | | | | Office-Home | | | | | VisDA | DomainNet-126 | | | | |
|---|---|---|---|---|---|---|---|---|---|---|---|---|---|---|---|---|
| | | →A | →D | →W | →Avg. | →Ar | →Cl | →Pr | →Rw | →Avg. | Sy→Re | →C | →P | →R | →S | →Avg. |
| OpenCLIP | CVPR23 | 85.7 | 91.2 | 91.8 | 89.6 | 83.8 | 76.1 | 93.5 | 92.3 | 86.4 | 86.7 | 86.4 | 82.0 | 92.3 | 80.8 | 85.4 |
| **ProDe** w/ OpenCLIP | – | **86.7** | **97.2** | **97.1** | **93.7** | **87.2** | **82.4** | **95.5** | **93.4** | **90.3** | **91.4** | **89.4** | **86.7** | **93.7** | **84.4** | **88.6** |

**Full results of generalized SFDA.** As a supplement to the average results of the generalized SFDA results (reported in Tab. 6), Tab. 14 presents the full results on 12 transfer tasks, including S-, T- and H-accuracy. In terms of H-accuracy, ProDe-V achieves the best results on half tasks. These results are not only due to significant improvements in the target domain (see T-accuracy) but also derive from a balanced drop in the source domain (see S-accuracy).

**Full results of TTA.** As a supplement to the average results of the TTA results (reported in Tab. 7), Tab. 15 presents the full results on the Office-Home dataset. On all 12 transfer tasks, ProDe-V achieves substantial increase, leading to **12.7**% gains on top of the second-best method T3A.

## E.2 EXPANDED MODEL ANALYSIS

**Reliance analysis on ViL models.** As illustrated in the right of Fig. 2, our proxy denoising is executed at the logit level, which means that the proposed method does not depend on a specific ViL model, such as CLIP, since it does not utilize the internal structure of these models. To validate this claim, we conduct an extensive test with OpenCLIP (Cherti et al., 2023). Meanwhile, we selected DIFO, the previous best ViL-based method, for comparison. Tab. 16∼19 present comparison results across all four datasets. Regardless of whether we use CLIP or OpenCLIP as the ViL model, ProDe beats DIFO in average accuracy. Furthermore, the relative gains are consistent. In comparison to DIFO, ProDe improves approximately by **0.3**%, **2.0**%, **1.0**% and **4.5**% on Office-31, Office-Home, VisDA and DomainNet-126, respectively. This trend suggests that our method is generic with the ViL model, and can readily benefit from the advancement in ViL models.

In addition, Tab. 20 displays a comparison of the zero-shot results from OpenCLIP. In all tasks (which are detailed in the "Full results of the comparison to CLIP's zero-shot" section of Sec.E.1), ProDe w/ OpenCLIP surpasses OpenCLIP. This suggests that the task-specific target model effectively incorporates generic knowledge in ViL models.

Table 21: Ablation results (%) of prompt learning on **Office-31**, **Office-Home** and **VisDA**.

| # | Method | Office-31 | Office-Home | VisDA | Avg. |
|---|---|---|---|---|---|
| 1 | **ProDe**-V w/o prompt | 91.7 | 81.9 | 88.0 | 87.2 |
| 2 | **ProDe**-V | **92.6** | **84.5** | **91.0** | **89.3** |

Table 22: Comparison of training resource demands (per iter.) on Ar→Cl in **Office-Home**.

| # | Item / Method | SHOT | AaD | **ProDe** |
|---|---|---|---|---|
| 1 | GPU memory consumption↓ (G) | **7.868** | 9.622 | 9.851 |
| 2 | Training times↓ (s) | **0.407** | 0.547 | 0.491 |

**Effect of prompt learning.** In ProDe, prompt learning contributes to knowledge synchronization. To isolate its effectiveness, we propose a variation method ProDe-V w/o prompt that removes prompt learning. As shown in Tab. 21, the absence of prompt learning results in **2.1**% decrease in average accuracy. These results indicate that this prompt learning might reduce the proxy error by tuning space $D_V$ close to the domain-invariant space $D_I$, meeting our expectations.

**Training resource demands.** To evaluate the training resource demands, we select two typical methods without using ViL model, SHOT and AaD, as comparisons. We conducted the test using the transfer task Ar → Cl from Office-Home, under the same testing conditions, including mini-batch size. The results, as shown in Tab. 22, indicate that despite using a large ViL model, our approach does not incur significant additional training costs and requires a similar amount of computational resources. This is because: (1) The ViL model is frozen in our method, making its use efficient, and (2) Our ProDe approach does not require a feature bank with periodic updates for deep clustering like SHOT, nor does it involve identifying neighborhoods as in AaD.

**Sensitivity of prompt initialization.**
In the proposed approach, we employ the initialization template of "a photo of a <cls>" for each class because it is the most used template to initiate the learnable prompt. The effect of prompt learning with this initiation is evaluated as reported in Tab. 21.

For further analysis, we conduct an ablation study on nine typical initialization templates. As shown in Tab. 23, there are no evident performance variations crossing the Office-31, Office-Home, and VisDA datasets, indicating

Table 23: Ablation study results (%) for typical prompt templates on **Office-31**, **Office-Home** and **VisDA**.

| # | Initialization template | Office-31 | Office-Home | VisDA |
|---|---|---|---|---|
| 1 | 'X [CLS].'(#X=4) | 91.9 | 84.2 | 90.5 |
| 2 | 'X [CLS].'(#X=16) | 91.9 | 82.9 | 90.5 |
| 3 | 'There is a [CLS].' | **93.0** | 83.1 | 90.6 |
| 4 | 'This is a photo of a [CLS].' | 92.4 | 83.2 | 90.7 |
| 5 | 'This is maybe a photo of a [CLS].' | 92.6 | 84.2 | **91.0** |
| 6 | 'This is almost a photo of a [CLS].' | 92.4 | 84.2 | 90.8 |
| 7 | 'This is definitely a photo of a [CLS].' | 92.5 | 84.2 | 90.7 |
| 8 | 'a picture of a [CLS].' | 92.2 | **84.5** | 90.7 |
| 9 | 'a photo of a [CLS].' | 92.6 | **84.5** | **91.0** |

that our method is insensitive to the selection of templates. Furthermore, the semantic templates outperform those that use 'X' (see rows 1 and 2). These results align with our expectations.

**Parameter sensitivity.** In this part, we discuss the parameter sensitivity of parameters $\alpha$, $\beta$ in $L_{ProDe}$ (see Eq. (6)) and correction strength parameter $\omega$ in proxy denoising (see Eq. (5)). All experiments are conducted based on the transfer tasks Ar→Cl in the Office-Home dataset. The varying range are set to $0.5 \leq \alpha \leq 1.4$, $0.1 \leq \beta \leq 1.0$ and $0.5 \leq \omega \leq 1.4$ in 0.1 step size. Fig. 5 (a) depicts the results as $\alpha$–$\beta$ vary. When the two parameters changes, there are no evident drops in the accuracy variation curves, except for two boundary situations: (1) $\alpha = 0.5$ and (2) $\beta = 1.0$. The results indicate that ProDe is insensitive to parameters $\alpha$ and $\beta$. Meanwhile, when we select parameters, $\alpha$'s value should be lager than $\beta$. Besides, in Fig. 5 (b) and (c), we display the results when $\alpha \times \omega$ and $\beta \times \omega$ vary, respectively. Thus, we present the relation between the correction strength and regularization elements in $L_{\mathrm{ProDe}}$. From the two sub-figures, it is seen that the performance has a significant drop as we adopt $\omega = 1.4$. This show that the correction strength in the proxy denoising block should not be too strong.

## F  LIMITATION AND FUTURE WORK

ProDe has shown impressive performance in multi-SFDA settings, highlighting its efficacy. However, it is important to note that it is specifically designed for a white-box scenario, which may not be applicable in certain real-world contexts. For the kind of black-box application, such as models in the cloud, our proxy denoising may not work well since all details of the model, including the

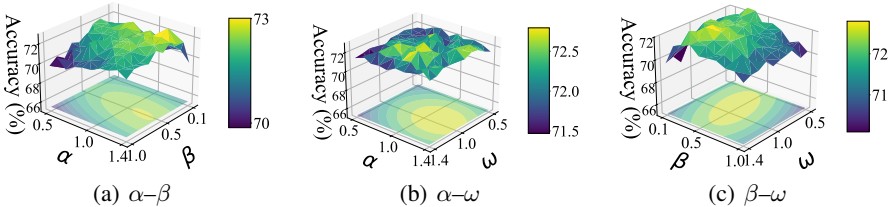

(a) $\alpha-\beta$      (b) $\alpha-\omega$      (c) $\beta-\omega$

Figure 5: Sensitivity analysis of hyper-parameters $\alpha$, $\beta$ and $\omega$.

required logits features, are transparent to us. In the future, finding ways to extend our method to this challenging scenario will be an interesting direction.

