# OpenReview forum: "Proxy Denoising for Source-Free Domain Adaptation"
_ICLR.cc/2025/Conference — ICLR 2025 Oral_

### Official Review · Reviewer_X5vf · 2024-10-22

**Soundness:** 3
**Presentation:** 4
**Contribution:** 3
**Rating:** 8
**Confidence:** 3

**Summary:**

This paper introduces Proxy Denoising (ProDe) to improve Source-Free Domain Adaptation (SFDA) by addressing noisy predictions from Vision-Language (ViL) models. The authors propose a proxy denoising mechanism based on proxy confidence theory to correct these noisy predictions and guide adaptation toward a domain-invariant space. They further enhance this process with mutual knowledge distillation regularization. Experiments demonstrate that ProDe outperforms existing methods across various SFDA settings, including closed-set, open-set, partial-set, and generalized scenarios.

**Strengths:**

*	The paper is well-written, well-organization, and easy to follow.
*	The insight on rectifies the inaccurate predictions of ViL models significantly contributes to SFDA settings.
*	The paper introduces a novel ProDe method, which effectively corrects ViL model predictions through the use of a proxy confidence theory, offering a reliable approach to prediction refinement.
*	The mutual knowledge distillation regularization is a strong addition, enabling the model to capitalize on refined proxy predictions with improved efficiency.
*	The authors evaluate the proposed method through extensive experiments, including challenging partial-set, open-set, and generalized SFDA settings, demonstrating the versatility of the method.

**Weaknesses:**

*	Overall, this paper is of high quality, with clear motivation and novel insights. The proposed ProDe method also demonstrates strong performance in the SFDA scenario. It would be interesting to see additional results in a similar scenario, such as test-time domain adaptation.

**Questions:**

*	Could the authors provide more detailed descriptions for Fig. 1? Additional explanations would help readers better understand the key ideas of the paper.
---
I would consider increasing my score if the authors address the concerns raised by me and other reviewers.

---

> ### Author Response · Authors · 2024-11-18
>
> Thank you so much for the great comments.  Our response to your concerns is presented as follows.
>
> $\textcolor{orange}{\text{Q1:}}$ **It would be interesting to see additional results in a similar scenario, such as test-time domain adaptation.**
>
>
> $\textcolor{green}{\text{Response:}}$ Great suggestion. We have evaluated the proposed method on the Office-Home dataset in the Test-Time Adaptation (TTA) setting. As shown in the table below, our method demonstrates advantages over previous state-of-the-art methods (results cited from [1], where all methods maintain a fixed batch size of 64, similar to ours). We will include these results in the revised manuscript to enhance our analysis.
>
>
> Table: Comparison results (%) in the TTA setting.
> | Method  | Ar→Cl | Ar→Pr | Ar→Rw | Cl→Ar | Cl→Pr | Cl→Rw  | P→Ar | Pr→Cl | Pr→Rw | Rw→Ar | Rw→Cl | Rw→Pr | Avg. |
> |:--------------|----:|:----:|:----:|:----:|:----:|:----:|:----:|:----:|:----:|:----:|:----:|:----:|:----:|
> | Tent [2]  | 47.6  | 63.2   | 72.3   | 57.1   | 63.7   | 65.9   | 55.9   | 46.6   | 72.7   | 67.7   | 51.8   | 77.1   | 61.7   |
> | T3A [3]   | 49.7  | 73.2   | 77.0   | 55.5   | 67.7   | 68.5   | 55.8   | 46.1   | 75.7   | 67.0   | 49.6   | 78.0   | 63.8   |
> | CoTTA [4] | 44.5  | 62.5   | 72.3   | 55.4   | 63.0   | 65.3   | 54.9   | 46.0   | 76.7   | 66.0   | 49.5   | 76.7   | 60.5   |
> | EATA [5]  | 46.4  | 62.5   | 72.2   | 55.3   | 65.8   | 65.8   | 53.8   | 43.4   | 76.4   | 66.5   | 50.5   | 76.4   | 60.7   |
> |SAR [6]   | 45.3  | 61.9   | 71.9   | 55.4   | 66.4   | 65.7   | 53.7   | 42.7   | 72.5   | 66.4   | 49.3   | 76.2   | 60.3   |
> | **ProDe-V (ours)**   | **64.5** | **84.9** |**84.7** | **76.1** | **85.1** | **83.7** | **75.5** | **64.0** | **85.1** | **77.4** | **67.3** | **87.1**| **78.0**|
>
>
> [1] Benchmarking test-time adaptation against distribution shifts in image classification. arXiv preprint arXiv:2307.03133, 2023.
>
> [2] Tent: Fully test-time adaptation by entropy minimization. In ICLR20.
>
> [3] Test-time classifier adjustment module for model-agnostic domain generalization. In NeurIPS21.
>
> [4] Continual test-time domain adaptation. In CVPR22.
>
> [5] Efficient test-time model adaptation without forgetting. In ICML22.
>
> [6] Towards stable test-time adaptation in dynamic wild world. In ICLR23.
>
>
> $\textcolor{orange}{\text{Q2:}}$ **Could the authors provide more detailed descriptions for Fig. 1? Additional explanations would help readers better understand the key ideas of the paper.**
>
> $\textcolor{green}{\text{Response:}}$ To address this issue, we will revise the caption of Fig. 1 as follows:
>
> “Conceptual illustration of ProDe. We align the adapting direction with the desired trajectory by leveraging a proxy space that approximates the latent domain-invariant space. This process incorporates direction adjustments based on proxy error correction, effectively implementing proxy denoising, and finally achieves enhanced model adaptation.”
>
> We believe this revision will clarify the key ideas presented in the figure and improve understanding for our readers.

---

> > ### Comment · Reviewer_X5vf · 2024-11-21
> >
> > Thank you for your efforts. My comments have been adequately addressed. I would encourage the authors to include additional comparisons related to TTA in the main paper and to release the source code upon the paper's acceptance. In light of this, I have decided to increase my score.

---

> > > ### Author Response · Authors · 2024-11-21
> > >
> > > Thank you for your feedback and consideration! We will add those TTA results in the revised manuscript.

---

### Official Review · Reviewer_SdQs · 2024-10-28

**Soundness:** 3
**Presentation:** 3
**Contribution:** 2
**Rating:** 8
**Confidence:** 4

**Summary:**

The authors tackle Source-Free Domain Adaptation (SFDA), where a pre-trained model adapts to an unlabeled target domain without access to source data. While Vision-Language (ViL) models show potential for SFDA, they often generate noisy predictions, an issue the authors investigate in this context. To address it, they propose Proxy Denoising (ProDe), a novel method that leverages proxy confidence theory to correct the ViL model’s predictions and introduces mutual knowledge distillation to make better use of these refined predictions. Extensive experiments on standard benchmarks demonstrate that ProDe outperforms prior methods across conventional closed-set, as well as partial-set, open-set, and generalized SFDA settings. The authors intend to release their code.

**Strengths:**

The paper effectively addresses the important problem of source-free domain adaptation (SFDA), where models must adapt to new target domains without access to labeled source data—an increasingly relevant setup in practical scenarios where source data may be proprietary or sensitive. Demonstrating state-of-the-art performance on standard SFDA benchmarks, the proposed method showcases its robustness and potential impact in the field. The authors employ mutual knowledge distillation to synchronize knowledge, reducing noise and enabling reliable knowledge transfer, and they incorporate category balance regularization to prevent “category collapse,” ensuring balanced treatment of each class. Through extensive analysis, including feature distribution visualizations and thorough ablation studies, the paper provides deep insights into model behavior and validates the effectiveness of each component. Furthermore, the paper is well-documented, with a clear presentation of the experimental setup, benchmark datasets, and evaluation metrics, enhancing reproducibility and accessibility.

**Weaknesses:**

### Ambiguity and Misleading Terminology in Domain Invariance Claims:
The authors claim that their method moves toward a “domain invariant space”  $D_v$ starting from  $D_t$ , but this terminology is misleading and theoretically problematic. If a domain-invariant space were achievable, there would be no need for adaptation across other domains, as all target domains would align seamlessly with this invariant space. However, the results suggest that domain shifts still impact the model and hence there is a need for adaptation for every domain, which contradicts the premise of invariance. For example, if the model were genuinely invariant after training on a source domain (e.g., Ar), it should perform equally well across all other domains (e.g., Cl, Pr, Rw) without further adaptation. This contradiction suggests that  $D_v$  is not genuinely invariant, but rather biased towards the target domain.

It would be beneficial if the authors could redefine the term “domain invariant space.” They might consider an alternative term, such as “target-aligned space,” which more accurately reflects the observed need for per-domain adaptations.

###	Unsupported Assumption Regarding  $e_{VI}$  and Invariant Space Error:
On line 164, the authors assert that  $e_{VI}$  could be ignored, implying that the error between the vision-language model’s space and the purported domain-invariant space is negligible. This assumption is dubious without further justification. The vision-language model’s embedding space may indeed diverge from the so-called invariant space, leading to substantial misalignment and error. Ignoring  $e_{VI}$  risks undermining the model’s robustness in handling domain shifts.

I suggest the authors provide empirical evidence for dismissing  $e_{VI}$ and validation of $d_I^0 \approx d_V^0 \gg e_{VI}$ by conducting experiments across a range of scenarios like different domain adaptation settings.

###	Oracle Configuration:
There is a flaw in the Oracle experiment setup, particularly in the Cl-to-Ar scenario (Line 410). The Oracle is incorrectly trained on the source domain (Cl) rather than the target domain (Ar). If this was a typo, I recommend correcting it. Otherwise, if this is intentional, it would be helpful to clarify the rationale for training on the source domain and claiming it to be an oracle.

###	Over-Reliance on a Single Vision-Language Model (CLIP):
The authors rely solely on CLIP for their experiments, neglecting to evaluate the method’s effectiveness with other vision-language models (e.g., LLaVA, Llama). This limitation raises concerns about the generalizability of the approach. Vision-language models have varied architectures and domain alignment properties, and the performance may vary significantly across models. Without results on other models, it is unclear if the proposed method is tailored specifically to CLIP or if it can generalize to other ViL models.

I suggest that the authors expand their experiments to include other vision-language models, such as LLaVA and Llama. Reporting these results would provide valuable insights into the generalizability of ViL models.

###	Generalization Claims to SFDA Settings:
The authors claim that their approach can generalize to broader SFDA settings, yet they do not provide any insights, or experimental results for critical scenarios like source-free multi-target domain adaptation (SF-MTDA [1])  and source-free multi-source domain adaptation (SF-MSDA [2]).

To enhance the rigor of their claims, I recommend that the authors either include experimental results for SF-MTDA and SF-MSDA scenarios or provide a detailed discussion on potential limitations or adaptations necessary for these settings. This addition would clarify the scope and limitations of the method’s applicability.


### References

[1] Kumar, Vikash, et al. "Conmix for source-free single and multi-target domain adaptation." Proceedings of the IEEE/CVF Winter Conference on Applications of Computer Vision. 2023.

[2] Ahmed,Miraj, et al. "Unsupervised Multi-Source Domain Adaptation Without Access to Source Data."  Proceedings of the IEEE/CVF Conference on Computer Vision and Pattern Recognition (CVPR), 2021.

**Questions:**

It would be great if the authors could clarify the doubts above especially those related to the theory and domain invariance space.

# UPDATE (After Discussion Period):
The discussion addresses all my concerns, and I appreciate the detailed responses and clarifications provided. Consequently, **I am increasing my ratings**. The responses demonstrate a strong understanding of the core issues, effectively addressing ambiguity, unsupported assumptions, and the generalization of claims. The inclusion of additional experiments, such as evaluations with OpenCLIP and broader SFDA settings, further solidifies the robustness and adaptability of the proposed method. The effort to provide comprehensive results, insightful explanations, and necessary corrections is commendable. Thank you for your diligence and thoroughness in addressing these points.

---

> ### Author Response · Authors · 2024-11-18
> **Author response**
>
> Thank you so much for the great comments.  Our response to your concerns is presented as follows.
>
> $\textcolor{orange}{\text{Q1:}}$ **Ambiguity and Misleading Terminology in Domain Invariance Claims**.
>
> $\textcolor{green}{\text{Response:}}$ Insightful discussion. In the context of domain adaptation, "domain invariant space" refers to an ideal latent embedding space where the mapped features from different domains align with the same probability distribution. A general goal of domain adaptation is to approach this ideal space, although achieving it perfectly in practice is often not feasible. This terminology is widely accepted within the domain adaptation community.
>
> While we understand the suggestion to redefine the term, introducing a new term is often conservative unless a fundamentally new concept is presented, which is not the case here. Therefore, we prefer to retain the term "domain invariance" while further elaborating on its meaning to enhance understanding. However, we are still open to more suggestions.
>
>
> $\textcolor{orange}{\text{Q2:}}$ **Unsupported Assumption Regarding and Invariant Space Error**.
>
> $\textcolor{green}{\text{Response:}}$ We appreciate your concern regarding the assumption. As noted in our analysis presented in Figure 4 (d) (Lines 510-519), our findings indicate that the impact of denoising $e_{VI}$ is negligible during the early phases of domain adaptation. That means, in this initial stage the divergence between the Vision-Language model’s space and the domain-invariant space does not significantly influence adaptation outcomes.
>
> We will ensure to clarify this in the revised manuscript.
>
>
> $\textcolor{orange}{\text{Q3:}}$ **Oracle Configuration**.
>
> $\textcolor{green}{\text{Response:}}$ Apologies for this typo, and we will correct it.
>
>
>
> $\textcolor{red}{\text{Q4:}}$ **Over-Reliance on a Single Vision-Language Model (CLIP)**.
>
> $\textcolor{green}{\text{Response:}}$ Our selection of CLIP is based on its widespread use in existing UDA and SFDA research, ensuring fair comparison. We recognize the importance of evaluating our method with other vision-language models and appreciate the suggestion to expand our analysis. We have conducted this test with OpenCLIP [1], selecting the previous best ViL method DIFO [2] as a comparison.
> The results in the table below indicate that the proposed method is generic with the ViL model, and can readily benefit from the advancement in ViL models.
>
> We will add the detailed results in the revised manuscript.
>
> Table: Reliance analysis on ViL models.
> | **Method**| **Office31** | **Office-Home** | **VisDA** | **DomainNet-126** |
> |-----|:-----:|:----:|:----:|:-----:|
> |DIFO w/ CLIP [2]| 92.5| 83.1| 90.3| 80.0|
> |**ProDe w/ CLIP**|**92.6**|**86.2**|**91.6**|**85.0**|
> |DIFO w/ OpenCLIP [2]|93.2|87.6|90.7| 84.6|
> | **ProDe w/ OpenCLIP**| **93.3**| **90.3**| **93.0**|**88.6**|
>
> [1] Reproducible scaling laws for contrastive language-image learning, In CVPR23.
>
> [2] Source-free domain adaptation with frozen multimodal foundation model, In CVPR24.
>
>
>
> $\textcolor{red}{\text{Q5:}}$ **Generalization Claims to SFDA Settings**.
>
> $\textcolor{green}{\text{Response:}}$ Thank you for your suggestion. We have evaluated ProDe in broader SFDA settings, specifically for source-free multi-target domain adaptation (SF-MTDA) and source-free multi-source domain adaptation (SF-MSDA). For SF-MTDA, we treat multiple target domains as a single integrated domain and adapt the source model accordingly. For SF-MSDA, we follow the ensembling approach from [1], passing the target data through each adapted source model and averaging the soft predictions to derive the test labels. The results, as shown in the table below, demonstrate that our method substantially outperforms state-of-the-art alternatives in both settings.
>
> We will include these findings in the revised manuscript.
>
> Table: SF-MTDA results (%) on Office-Home
> | **Method** | **Ar** | **Cl** | **Pr** | **Rw** | **Avg.** |
> | ----------------- |:---------:|:---------:|:---------:|:----------:|:--------:|
> | CoNMix [2]             | 75.6  | 81.4  | 71.4   | 73.4   | 75.4    |
> | **ProDe-V (ours)**      | **84.4**   | **89.4** | **80.9**  | **81.9** | **84.2** |
>
> Table: SF-MSDA results (%) on Office-Home
> | **Method** | **Ar,Cl,Pr→ Rw** | **Ar,Cl,Rw→ Pr** | **Ar,Pr,Rw→ Cl** | **Cl,Pr,Rw→ Ar** | **Avg.** |
> | ----------------- |:---------:|:---------:|:---------:|:----------:|:--------:|
> | SHOT-Ens [3]      | 82.9      | 82.8   | 59.3     | 72.2     | 74.3    |
> | DECISION [1]      | 83.6      | 84.4   | 59.4     | 74.5      | 75.5   |
> | **ProDe-V-Ens (ours)**  | **92.8**   | **93.8** | **75.5**  | **85.3**  | **86.8** |
>
>
> [1] Unsupervised multi-source domain adaptation without access to source data, In CVPR21.
>
> [2] Conmix for source-free single and multi-target domain adaptation, In WACV23.
>
> [3] Do we really need to access the source data? source hypothesis transfer for unsupervised domain adaptation, In ICML20.

---

> > ### Author Response · Authors · 2024-11-25
> >
> > Dear Reviewer SdQs,
> >
> > Thanks again for the valuable comments and suggestions. As the discussion phase is nearing its end, we wondered if the reviewer might still have any concerns that we could address. We believe our point-by-point responses addressed all the questions/concerns.
> >
> > It would be great if the reviewer could kindly check our responses and provide feedback with further questions/concerns (if any). We would be more than happy to address them. Thank you！
> >
> > Best regards Paper 5075 Authors.

---

### Official Review · Reviewer_hEmB · 2024-10-31

**Soundness:** 3
**Presentation:** 3
**Contribution:** 3
**Rating:** 8
**Confidence:** 4

**Summary:**

The previous methods using ViL for pseudo-supervision can generate noise, which introduces negative effects that have been overlooked. This paper proposes a ProDe method that first introduces a proxy confidence theory, which vividly analyzes and explains the sources of noise in ViL predictions, and specifically designs a denoising mechanism to correct ViL's predictions. Additionally, it introduces a mutual information extraction method to achieve knowledge synchronization between the ViL model and the target model.

**Strengths:**

The paper is based on proxy confidence theory and designs a reliable denoising algorithm to reduce the prediction noise of ViL, addressing an important issue that has been neglected in the use of ViL for pseudo-supervision. This may facilitate subsequent related work.

**Weaknesses:**

Although the ViL model is obtained based on a large dataset, for specific source and target domains, the ViL model can approximate the domain-invariant space. The validity of this assumption requires further theoretical support.

**Questions:**

I would like to know if using ViL for pseudo-supervision is suboptimal when the distance between the source domain space D_S (or training dataset D_Tt) and D_I is closer than the distance between D_V and D_I. Additionally, the formulas (5) and (6) in the paper contain many hyperparameters; is tuning these hyperparameters a challenge?

---

> ### Author Response · Authors · 2024-11-18
>
> Thank you so much for the great comments.  Our response to your concerns is presented as follows.
>
>
> $\textcolor{orange}{\text{Q1:}} $ **I would like to know if using ViL for pseudo-supervision is suboptimal when the distance between the source domain space $D_S$ (or training dataset $D_{Tt}$) and $D_I$ is closer than the distance between $D_V$ and $D_I$. Additionally, the formulas (5) and (6) in the paper contain many hyperparameters; is tuning these hyperparameters a challenge?**
>
> $\textcolor{green}{\text{Response:}}$ Thank you for this insightful question. We believe that using ViL for pseudo-supervision, despite potential distance discrepancies between the domain spaces, still provides valuable generic knowledge throughout the adaptation process.
> In such cases as mentioned, our proposed Mutual Knowledge Distillation method plays a crucial role. It effectively mitigates potential negative effects by integrating task-specific knowledge from the in-training target model with the generic knowledge from the ViL model. This integration ensures that our approach is not overly reliant on the ViL model alone, allowing it to adapt more effectively to the nuances of the target domain.
>
> Regarding the hyperparameters in Equations (5) and (6), we have provided their values used in our experiments in the "Hyperparameter Setting" section of Appendix-D. Specifically, we set the values of $(\alpha, \beta, \omega) $ to (1, 1, 0.4) consistently across all datasets, which did not require extensive fine-tuning. We have also discussed their insensitivity in Appendix E.2 (see Fig. 5). The parameter $\gamma$ is the only one that necessitates fine-tuning, as it is sensitive to the dataset scale, also noted in the TPDS method [1].
>
> [1] Source-free domain adaptation via target prediction distribution searching. International Journal of Computer Vision (IJCV), 132(3):654–672, 2024.

---

### Official Review · Reviewer_vhPY · 2024-11-07

**Soundness:** 3
**Presentation:** 3
**Contribution:** 3
**Rating:** 8
**Confidence:** 4

**Summary:**

The paper addresses Source-Free Domain Adaptation (SFDA) in terms of utilizing Vision-Language Models (VLMs) for supervision. Specifically, the authors argue that prior works that utilize VLMs for SFDA treat their predictions as the ground truth without considering the potential noise in their predictions. To alleviate this issue, the authors propose Proxy Denoising (ProDe)—an SFDA framework that corrects the supervisory VLM’s predictions before target adaptation. Extensive experiments on various Domain Adaptation benchmarks demonstrate the effectiveness of the approach compared to prior works in multiple domain adaptation settings. Moreover, analysis experiments substantiate the intuition behind the proposed method.

**Strengths:**

- **Presentation-** The paper is written well overall and conveys the central ideas of the work quite effectively. The paper is easy to follow and understand. Additionally, the authors have presented experiments on a wide range of benchmarks and settings.

- **Novelty-** The authors claim that the paper is the first to analyze the inaccurate predictions of the teacher VLM in the context of SFDA and propose a method to alleviate the same.

- **Results-** The authors present extensive experiments across several domain adaptation benchmarks and comparisons with several prior works that do and do not use VLMs for training (although whether some of the comparisons are fair is a question, see Weaknesses for details).

**Weaknesses:**

### (a) Concerns with Proxy Confidence Theory
- Theorem 1 provides a relation between the confidence of the VLM’s predictions and the confidence of the source model and the current training model. This is based on the approximation of the VLM’s predictions to a Gaussian distribution and further expressing this in terms of the confidence of the VLM’s predictions.
- However, the intuition behind how a Gaussian distribution is considered for the VLM’s predictions is not entirely clear. Moreover, the conversion in Eq. 2 also seems unclear, in terms of how the conversion is possible. Essentially, it would be better if the authors could explain L185-188 in more detail.

### (b) Fairness of comparisons
- The authors present comparisons of their proposed method ProDe with prior SFDA works and with works utilizing VLMs. Based on the implementation details provided in the supplementary, it appears that the prior SFDA works that do not utilize VLMs make use of ResNet-50 or ResNet-101 depending on the difficulty of the dataset.
- Can these comparisons of ProDe with prior SFDA works be considered fair? ProDe uses supervision from a VLM that has been pre-trained on WiT-400M while the SFDA works consider an ImageNet pre-trained ResNet-50 or ResNet-101. There is a massive difference in the models being used for adaptation. Although the student model is a vision-only backbone in ProDe, it is supervised by a VLM during target adaptation.
- The authors need to discuss these differences in the experiment settings to provide a more complete picture of the results. Additionally, the authors should present the comparisons in a fair setting, i.e., similar supervisory signals or backbones should be used in both the proposed method and the prior works.

**Questions:**

- How does the proposed method ProDe perform in a multi-source or multi-target domain adaptation setting? Can the authors present these results on OfficeHome or DomainNet?
- Why have the authors used DomainNet-126 rather than the full DomainNet dataset? Given that DomainNet is the most challenging domain adaptation benchmark among the chosen datasets, it is an important result.
- Following the discussion on fair comparisons in the previous section, can the authors present results in a fair setting? Eg. the authors can present results of prior SFDA works by using the CLIP vision encoder as the initialization rather than the ImageNet pre-trained backbones. Another possibility could be a baseline that provides VLM supervision to prior SFDA works.
- Additionally, if the above is not possible, could the authors present results with ViTs rather than ResNet-50 / ResNet-101 for a more fair comparison? Given that the results with VLMs use ViT-B, the prior works should use a backbone with a similar capacity.

---

> ### Author Response · Authors · 2024-11-18
>
> Thank you very much for the great comments. Our response to the your questions are elaborated below.
>
> $\textcolor{orange}{\text{Q1:}} $ **The intuition behind how a Gaussian distribution is considered for the VLM’s predictions is not entirely clear. Moreover, the conversion in Eq. 2 also seems unclear, in terms of how the conversion is possible.**
>
> $\textcolor{green}{\text{Response:}}$ Thank you for your insightful question regarding Theorem 1 and the treatment of the VLM's predictions as a Gaussian distribution. This assumption stems from the Central Limit Theorem, which suggests that, under certain conditions, the sum of a large number of independent random variables will tend to be distributed normally, regardless of the original distributions of the variables. In our context, we consider the VLM’s predictions to be influenced by various sources of noise and uncertainty, which justifies the Gaussian approximation.
>
> Regarding the conversion in Eq. 2 (also see Lines 195-204), we express this relationship in terms of probability distributions to facilitate the understanding of how the confidence of the VLM's predictions relates to the current training model and the source model. By framing the prediction as a probabilistic event, we can leverage the concept of proxy confidence, $P(GP(V)=True,t)$, to quantify how reliable we consider the VLM’s predictions to be at any point in the adaptation process. In essence, this conversion allows us to connect the notion of prediction reliability with the underlying distributions, making it easier to reason about the impact of proxy errors and their effect on the adaptation process.
>
> We will further clarify these points.
>
>
> $\textcolor{orange}{\text{Q2:}} $ **How does the proposed method ProDe perform in a multi-source or multi-target domain adaptation setting?**
>
> $\textcolor{green}{\text{Response:}}$ Please take the responses to $\textcolor{red}{\text{Q5}} $ of reviewer **SdQs**.
>
>
>
> $\textcolor{orange}{\text{Q3:}} $ **Why have the authors used DomainNet-126 rather than the full DomainNet dataset?**
>
> $\textcolor{green}{\text{Response:}}$ This is to ensure fair comparison with existing works
> as this version DomainNet-126 has been extensively used with cleaned labels
> as compared to the original version (see Appdendix-C).
>
> We will clarify.
>
>
> $\textcolor{orange}{\text{Q4 and Q5:}}$ **Following the discussion on fair comparisons in the previous section, can the authors present results in a fair setting? Additionally, if the above is not possible, could the authors present results with ViTs rather than ResNet for a more fair comparison?**
>
>
> $\textcolor{green}{\text{Response:}}$ Per suggestion, we present comparisons with typical SFDA methods using ViT backbones (cited from DPC [1]), employing ViT-B/16. The results in the table below show that ProDe-V16 consistently outperforms DPC in most cases. An exception is that ProDe-V16 is only 0.8\% behind on Office-31, which may be attributed to potential overfitting on this relatively small dataset. Notably, even with a ResNet backbone for the target model, ProDe-V16 still surpasses DPC, which utilizes a ViT. Generally, using a ViT for such a small training dataset is unnecessary due to the tendency for overfitting.
>
> We will add this test.
>
> Table: Comparison results (%) on Office-31, Office-Home, VisDA and DomainNet-126.
> | **Method** | **VLM** | **Office-31** | **Office-Home** | **VisDA** | **DomainNet-126** |
> | ----------------- |:---------:|:---------:|:---------:|:----------:|:--------:|
> | SHOT-ViT [2]   | X   |91.4   |78.1   |--    |71.4  |
> | DIPE-ViT [3]    | X   |90.5   |78.2   |--    |--   |
> | DSiT-ViT [4]    | X   |93.0   |80.5   |--    |--   |
> | AaD-ViT [5]    | X   |--     |--     |--   |72.7  |
> | DPC [1]        |√   |**93.3**   |85.4   |--    |85.6  |
> | **ProDe-V16 (ours)** |√   |92.5 | **88.0** | **92.0** |**88.1** |
>
> [1] Towards Dynamic-Prompting Collaboration for Source-Free Domain Adaptation, In IJCAI24.
>
> [2] Do we really need to access the source data? source hypothesis transfer for unsupervised domain adaptation. In ICML20.
>
> [3] Exploring domain-invariant parameters for source free domain adaptation, In CVPR22.
>
> [4] Domain-specificity inducing transformers for source-free domain adaptation. In ICCV23.
>
> [5] Attracting and dispersing: A simple approach for source-free domain adaptation, In NeurIPS22.

---

> > ### Comment · Reviewer_vhPY · 2024-11-23
> > **Official Comment by Reviewer vhPY**
> >
> > Thanks for the authors’ response. They have adequately addressed most of my concerns. Based on the authors’ response and the reviews from the other reviewers, I have a few follow-up comments:
> >
> > - The proposed method ProDe utilizes learnable prompts $U$ (Fig. 2 right) that are passed along with the template text prompts “a photo of a <cls>”. However, there is no discussion on this aspect of the method. Why are these prompts being used? How many prompts are being used? How essential are the learnable prompts to ProDe?  Can the authors present an ablation study on these learnable prompts in addition to addressing the above queries?
> >
> > - Regarding the question from ***Reviewer SdQs*** about the proposed method's overreliance on CLIP, the reviewer believes that ProDe can work with any discriminative Vision-Language Model (VLM), i.e., a VLM that generates embeddings from the vision and text encoders as outputs, such as CLIP or ALIGN. The authors can present results with ALIGN, FILIP, or BLIP to support this claim.

---

> ### Author Response · Authors · 2024-11-23
>
> Thanks again for the valuable comments and suggestions. Our response to these new comments are elaborated below.
>
> $\textcolor{orange}{\text{Q6}}$: Insightful comment. The proposed method ProDe utilizes learnable prompts (Fig. 2 right) that are passed along with the template text prompts “a photo of a <cls>”. However, there is no discussion on this aspect of the method. Why are these prompts being used? How many prompts are being used? How essential are the learnable prompts to ProDe? Can the authors present an ablation study on these learnable prompts in addition to addressing the above queries?
>
> $\textcolor{green}{\text{Response}}$: In the proposed approach, we employ the initialization template of “a photo of a <cls>” for each class because it is the most used template to initiate the learnable prompt. In addition, we have evaluated the effect of prompt learning with this initiation (see Table 20 in the revised manuscript).
>
> For further analysis, we conduct an ablation study on nine typical templates. As shown in the table below, there are no evident performance variations, indicating our method is insensitive to the selection of templates.
>
> We have added the results in the revised manuscript.
>
> Table: Ablation study results on initialization template selection.
> |# |**Initialization template**|**Office-31**|**Office-Home**|**VisDA**|
> |:-|:-|:-:|:-:|:-:|
> |1 |'X [CLS].'(#X=4)|91.2|85.9|90.4|
> |2 |'X [CLS].'(#X=16)|90.9|85.4|90.8|
> |3 |'There is a [CLS].'|91.9|85.9|91.4|
> |4 |'This is a photo of a [CLS].'|92.3|86.0|91.4|
> |5 |'This is maybe a photo of a [CLS].'|92.6|86.1|**91.6**|
> |6 |'This is almost a photo of a [CLS].'|**92.7**|86.1|91.5|
> |7 |'This is definitely a photo of a [CLS].'|92.6|86.1|**91.6**|
> |8 |'a picture of a [CLS].'|**92.7**|**86.2**|**91.6**|
> |9 |'a photo of a [CLS].'|92.6|**86.2**|**91.6**|
>
>
>
>
> $\textcolor{orange}{\text{Q7}}$: Regarding the question from Reviewer SdQs about the proposed method's overreliance on CLIP, the reviewer believes that ProDe can work with any discriminative Vision-Language Model (VLM), i.e., a VLM that generates embeddings from the vision and text encoders as outputs, such as CLIP or ALIGN. The authors can present results with ALIGN, FILIP, or BLIP to support this claim.
>
>
> $\textcolor{green}{\text{Response}}$: Great suggestion. In response to **reviewer SdQs**'s concern about the overreliance on CLIP, we have further tested the generality of our method with OpenCLIP [1] as the ViL model. Prefere refer to the responses to $\textcolor{red}{\text{Q4}}$ of **reviewer SdQs** for more details. Additionally, the revised paper includes a thorough analysis in Tables 15–19 and the section "Reliance Analysis on ViL Models" in the supplementary document. We can test a third ViL model for the final version if needed.
>
> [1] Reproducible scaling laws for contrastive language-image learning, In CVPR23.

---

> > ### Author Response · Authors · 2024-11-25
> >
> > Dear Reviewer vhPY,
> >
> > Thanks again for the valuable comments and suggestions. As the discussion phase is nearing its end, we wondered if the reviewer might still have any concerns that we could address. We believe our point-by-point responses addressed all the questions/concerns.
> >
> > It would be great if the reviewer could kindly check our responses and provide feedback with further questions/concerns (if any). We would be more than happy to address them. Thank you！
> >
> > Best regards Paper 5075 Authors.

---

> > > ### Comment · Reviewer_vhPY · 2024-11-25
> > > **Official Comment by Reviewer vhPY**
> > >
> > > Thanks for the authors' response. They have adequately addressed my concerns, so I raise my score. I suggest the authors incorporate all the changes and the feedback from the other reviewers into the final manuscript.

---

> > > > ### Author Response · Authors · 2024-11-26
> > > >
> > > > Thank you for your feedback and consideration! We will incorporate all the changes and the feedback from the other reviewers into the final manuscript.

---

### Author Response · Authors · 2024-11-22

Per the reviewers' suggestions, we have revised the paper in two aspects (all modifications are marked in $\textcolor{blue}{\text{blue}}$ color to easy tracking and locating).

   1. Elaborate the conceptual illustration of ProDe (see the caption of Figure 1 in the revised manuscript, corresponding to $\textcolor{orange}{\text{Q2}}$ of **X5vf**);
   2. Explain why adopting Gaussian distribution to approximate the VLM’s predictions (see Lines 186--189 in the revised manuscript, corresponding to $\textcolor{orange}{\text{Q1}}$ of **vhPY**);
   3. Explain why performing the conversion presented in Eq. (2) (see Lines 198--202 in the revised manuscript, corresponding to $\textcolor{orange}{\text{Q1}}$ of **vhPY**);
   4. Elaborate the term of "domain invariant space" (see footnote at the bottom of Page 2 in the revised manuscript, corresponding to $\textcolor{orange}{\text{Q1}}$ of **SdQs**);
   5. Elaborate the empirical evidence for our assumption that the impact of denoising $e_{VI}$ is negligible at the early phase of domain adaptation (see Lines 516--519 in the revised manuscript, corresponding to $\textcolor{orange}{\text{Q2}}$ of **SdQs**).



* In terms of Experiments, the revision includes:

   1. Further evaluate the three suggested settings: SF-MTDA, SF-MSDA, and TTA (see Table 7 and section "Comparison on SF-MTDA, SF-MSDA and TTA settings" in the revised manuscript, corresponding to $\textcolor{orange}{\text{Q2}}$ of **vhPY** and $\textcolor{red}{\text{Q5}}$ of **SdQs**);
   2. The full results of TTA results (see Table 14 in the supplementary document, corresponding to $\textcolor{orange}{\text{Q1}}$ of **X5vf**);
   3. Further test the generality of our method with OpenCLIP as the ViL model (see Table 15--19 and section "Reliance analysis on ViL models" in the supplementary document, corresponding to $\textcolor{orange}{\text{Q4}}$ of **SdQs**);
   4. Further compare with SFDA methods using ViT-B/16 architecture (see Table 22 and section "Comparison with SFDA methods with ViT backbone" in the supplementary document, corresponding to $\textcolor{orange}{\text{Q4 and Q5}}$ of **vhPY**);
   5. Elaborate the selection of those trade-off parameters (see section "Hyper-parameter setting" in the supplementary document, corresponding to $\textcolor{orange}{\text{Q1}}$ of **hEmB**);
   6. Correct the typo in Oracle configuration (see Line 419 in the revised manuscript, corresponding to $\textcolor{orange}{\text{Q3}}$ of **SdQs**);
  7. Further ablation study on the prompt initialization (see Table 23 and the section  "Sensitivity of prompt initialization" in the supplementary document, corresponding to $\textcolor{orange}{\text{Q6}}$ of **vhPY**).

---

### Public Comment · ~Huayu_Mai1 · 2025-06-25
**A point for discussion regarding Equation (1)**

Dear Authors,

Thank you for your interesting work. I would like to bring attention to a point regarding the interpretation of Equation (1) and its implications for $\eta_t$.

The paper states that as training progresses, the quantity $|e_{VI}|/|d_v^t|$ in Equation (1) gradually increases, leading to an increase in the impact degree $\eta_t$. However, Equation (1) just establishes an upper bound for $\eta_t$:

$$\eta_t = \frac{|d_I^t|}{|d_v^t|} \le 1 + \frac{|e_{VI}|}{|d_v^t|}$$

While it is true that the term $1 + \frac{|e_{VI}|}{|d_v^t|}$ (the upper bound) increases with training, an increase in an upper bound does not logically imply that the value itself ($\eta_t$) is increasing. This conclusion drawn that $\eta_t$ increases solely based on the increase of its upper bound appears to be a logical fallacy. Further justification on a different line of reasoning would be necessary.

Thank you for your time and consideration.

---

> ### Public Comment · ~Song_Tang5 · 2025-06-25
>
> Thank you for your insightful comment and for drawing our attention to this issue.
>
> To begin with, Source-Free Domain Adaptation (SFDA) inherently involves uncertainty arising from unknown noise. Accordingly, Equation (1) can be interpreted through the lens of Distributionally Robust Optimization (DRO) theory, where the upper bound is understood as a worst-case estimate. Under this interpretation, the conclusion that $\eta_t$ gradually increases remains valid within the DRO framework, which takes uncertainty into consideration.
>
> We will revise the ArXiv version to clarify this point more explicitly.

---

### Meta-Review · Area_Chair_b7dj · 2024-12-18

**Metareview:**

This paper was reviewed by four experts in the field. Originally it got mixed ratings. During the discussion period, the authors successfully addressed reviewer's concerns. All reviewers gave a final rate of 8 after the discussion period. Reviewers agree that the paper is well written. It proposes a novel approach for source-free domain adaptation with extensive experiments. Overall, it is a solid work for ICLR.

**Additional Comments On Reviewer Discussion:**

Originally, reviewers raised some issues regarding fair comparison and several other issues (terminology, assumption, claims, etc in the paper). However, the authors successfully addressed those concerns during the discussion period. All reviewers raised their ratings to 8 in the end.

---

### Decision · Program_Chairs · 2025-01-22

Accept (Oral)